# Heat, salt, and volume transports in the eastern Eurasian Basin of the Arctic Ocean, from two years of mooring observations

Andrey Pnyushkov[1], Igor Polyakov[2], Robert Rember[1], Vladimir Ivanov[4, 6], Matthew B. Alkire[3], Igor Ashik[4], Till M. Baumann[1], Genrikh Alekseev[4], and Arild Sundfjord[5]

1 International Arctic Research Center, University of Alaska Fairbanks, Fairbanks, AK, USA

2 International Arctic Research Center and College of Natural Science and Mathematics, University of Alaska Fairbanks, AK, USA

3 Applied Physics Laboratory, University of Washington, Seattle, WA, USA

4 Arctic and Antarctic Research Institute, St. Petersburg, Russia

5 Norwegian Polar Institute, Fram Centre, Tromsø, Norway

6 Moscow State University, Geography department, Moscow, Russia

*Correspondence to*: Andrey Pnyushkov (avpnyushkov@alaska.edu)

**Abstract.** This study discusses along-slope volume, heat, and salt transports derived from observations collected in 2013–15 using a cross-slope array of six moorings ranging from 250 m to 3900 m in the eastern Eurasian Basin (EB) of the Arctic Ocean. These observations demonstrate that in the upper 780 m layer, the along-slope boundary current advected, on average, $5.1 \pm 0.1$ Sv of water, predominantly in the eastward (shallow-to-right) direction. Monthly net volume transports across the Laptev Sea slope vary widely, from $\sim 0.3 \pm 0.8$ in April 2014 to $\sim 9.9 \pm 0.8$ Sv in June 2014. $3.1 \pm 0.1$ Sv (or 60 %) of the net transport was associated with warm and salty intermediate-depth Atlantic Water (AW). Calculated heat transport for 2013-15 (relative to -1.8 °C) was $46.0 \pm 1.7$ TW, and net salt transport (relative to zero salinity) was $172 \pm 6$ Mkg/s. Estimates for AW heat and salt transports were $32.7 \pm 1.3$ TW (71 % of net heat transport) and $112 \pm 4$ Mkg/s (65 % of net salt transport). The variability of currents explains $\sim 90$ % of the variability of the heat and salt transports. The remaining $\sim 10$ % is controlled by temperature and salinity anomalies together with temporal variability of the AW layer thickness. The annual mean volume transports decreased by 25% from $5.8 \pm 0.2$ Sv in 2013–14 to $4.4 \pm 0.2$ Sv in 2014–15 suggesting that changes of the transports at interannual and longer time scales in the eastern EB may be significant.

## 1 Introduction

Propagating along the deep basin margins of the Arctic Ocean, the Arctic Circumpolar Boundary Current (ACBC; Aksenov et al., 2011) carries substantial amounts of Atlantic-origin heat and salt (Aagaard and Greisman, 1975; Schauer et al., 2008). The role of Atlantic Water (AW) heat advected by the ACBC in recent sea ice loss and thermohaline changes in the Eurasian Basin (EB) of the Arctic Ocean is still under debate (e.g., Carmack et al., 2015 for discussion). However, the AW heat may be important for the state of sea ice in regions where heat can be released to the bottom of sea ice by winter ventilation (Onarheim et al., 2014; Ivanov et al., 2016; Polyakov et al., 2017). Thus, quantifying ACBC transports is essential for understanding the current and future state of the Arctic Ocean and sea ice.

Warm and salty, intermediate-depth (~150-800 m) AW penetrates into the Arctic Ocean through two major gateways. The Fram Strait branch of AW enters the EB with the West Spitsbergen Current (WSC) through the eastern part of Fram Strait— the passage between Spitsbergen and the Greenland shelf (**Fig. 1**). The second, Barents Sea AW branch enters the Arctic Ocean through the Barents Sea Opening, and then flows into the EB through the St. Anna Trough. North of the Kara Sea, these branches merge laterally and propagate further along the EB slope in the shallow-to-right direction as two distinct flows, forming a confluence area along the shelf slope between the St. Anna Trough and the Lomonosov Ridge with a distinctive thermohaline front between these two streams (Schauer et al., 2002; Baumann et al., 2018). After the confluence, the Barents Sea AW branch flows eastward over the upper EB slope, while the Fram Strait branch occupies a broad segment of the lower slope.

Despite the important role the ACBC plays for the heat and salt balance of the EB, trustworthy observational estimates for heat, salt, and volume transports are rare. For the eastern EB and, in particular, for the central Laptev Sea slope, to our knowledge, such estimates do not yet exist. However, observations made during the period of 2013–15 by an array of six moorings crossing the continental slope of the Laptev Sea along the 125°E meridian (**Fig. 2**) were used in this study to provide the first estimates of volume, heat, and salt transports carried by the ACBC in the eastern EB (**Fig. 1**). These mooring records of water temperature, salinity, and current velocities were collected under the auspice of the Nansen and Amundsen Basins Observational System (NABOS) program. We describe here the temporal variability of these transports at scales resolved by the available mooring records. This study is focused on transports in the upper 780-m layer (limited by the available observations), with a particular focus on the AW, which is the major constituent of the ACBC transports at the Laptev Sea slope.

## 2 Background information

### 2.1 Water mass structure in the eastern EB and Laptev Sea

Climatological temperature and salinity profiles enable insight regarding the vertical structure of the water column at the Laptev Sea slope. Here we use the Polar Hydrographic Climatology (PHC) data set compiled for the 1970–90s (Steele at al.,

2001). According to the PHC temperatures and salinities averaged over the area 77–79ºN and 125-126ºE, the upper ~30 m layer of the water column in summer and ~50 m in winter is occupied by relatively fresh (S < 34) and well-mixed waters forming the surface mixed layer (SML, **Fig. 2** bottom panels). The thickness of the SML and its temperature and salinity vary across the slope and experience substantial seasonal changes due to interaction with the polar atmosphere, entrainment of deep water, ice formation and melting (e.g,. Baumann et al., 2018). In the Cold Halocline Layer (CHL; 30–100 m depth

range), which underlies the SML, salinity increases with depth, while temperature is still close to the freezing point (**Fig. 2**; Aagaard et al., 1981; Rudels et al., 1996). A strong density stratification in this layer due to salinity gradients suppresses the vertical exchange of heat and salt with the relatively warm (T > 0 °C) AW below (Aagaard and Greisman 1975; Aagaard et al., 1981; Rudels et al., 1996).

Between the CHL and the AW temperature core, both temperature and salinity increase with depth, forming the low

halocline layer (e.g., Alkire et al., 2017). The AW at the central Laptev Sea slope occupies the intermediate layer (typically below 150 m; Dmitrenko et al., 2006). According to the PHC dataset, the Fram Strait branch of the AW spans the layer between ~180 and ~750 m (determined by the position of 0 °C isotherm), but for some years, the lower AW boundary determined using measured conductivity-temperature-depth (CTD) profiles can be found significantly deeper, at ~1000–1200 m (see, for example, Fig. 7 in Pnyushkov et al., 2015).

Climatological temperatures and salinities suggest that, at the Laptev Sea slope, the Barents Sea branch of AW underlies the Fram Strait branch (**Fig. 2**). Before reaching the Laptev Sea slope, the Barents Sea branch of the AW experiences strong thermal transformation, caused by interaction with the cold atmosphere, local waters, and sea ice. At the exit from the Barents Sea, this water has predominantly negative temperatures and $27.8 < \sigma < 28.0$ kg/m$^3$ density range (e.g., Pfirman et al., 1994; Schauer et al., 1997, 2002; Lien and Trofimov, 2013) and propagates along the EB slope at depths below 700-

800 m.

### 2.2 AW transports

ACBC transport over the Laptev Sea slope is influenced by inflows from both the Fram Strait and the Barents Sea AW branches. Providing a large influx of water and potential vorticity from the Nordic and the Barents seas, these inflows change the large-scale pattern of sea level in the EB (e.g., Karcher et al. (2007) and Aksenov et al. (2011)), and thus, serve as

remote forcing for transport changes to the Laptev Sea region. The long-term monitoring of AW inflow through Fram Strait

was established in 1997 (Fahrbach et al., 2001; Schauer et al., 2004). Following Beszczynska-Möller et al. (2012), this inflow consists of several types of AW: the warm AW (water warmer than 2 °C), which propagates along the slope in the upper 400-m layer as a continuation of the Norwegian Atlantic Current, and the modified returning AW. Mooring-based observations carried out between 1997 and 2010 suggest a net (for all waters) transport by the WSC as high as 6.6 ± 0.4 Sv (here and further throughout the text, values after plus/minus signs indicate the standard error of the mean except for the correlation coefficients where they indicate the range of 95% confident interval). In Fram Strait, the WSC consists of three branches — the WSC core branch, the offshore branch, and the WSC recirculation branch. The overall transport of warm AW toward the Arctic Ocean for these three branches is approximately 3.0 ± 0.2 Sv (**Table 1**). About 1.3 ± 0.1 Sv of the AW transport is carried by the topographically-steered core branch of the WSC (annual mean transports vary in the range of 1.0–1.5 Sv). North of Fram Strait, this branch turns to the east and follows along the Svalbard northern slope further into the Arctic Ocean (see **Fig. 1** for details). About 1.7 ± 0.1 Sv of the AW transport in Fram Strait is carried by the highly variable (annual mean transports vary in the range of 1–2 Sv) offshore WSC branch—the branch which feeds the flow toward the Yermak Plateau. The large difference between the total transport in the West Spitsbergen Current and net AW inflow into the Arctic Ocean suggests substantial recirculation of the AW in the strait and in the surrounding region (Schauer et al., 2008; Beszczynska-Möller et al., 2012; Hattermann et al., 2016). Multi-year means of the AW inflow (T>1.5 °C; allowing for some heat loss between the mooring line in Fram Strait and the Yermak Plateau) crossing Yermak Plateau estimated from an operational model system are in agreement with the values given above (Koenig et al., 2017).

Several snapshot-based estimates of volume transports are available for the slope area of Svalbard. Despite the very different time scales of snapshot-based and long-term estimates of transports, the former provide useful information about the possible ranges in transport variability. For example, Våge et al. (2016)—using geostrophic velocities at two CTD cross-sections across the boundary current flow near 30°E in 2012—reported a net AW transport of 1.6 ± 0.3 Sv (**Table 1**). The authors found evidence of a large warm-core (anti-cyclonic) eddy north of the main core of the boundary current potentially affecting the mean volume transport calculations.

All above estimates of transports are sensitive to the definition of the AW. Based on multiple cross-slope CTD sections in the area between 21°E and 33°E in 2013, Pérez-Hernández et al. (2017)—using density- and salinity-based criteria defining the AW ($27.7 < \sigma < 27.97$ kg/m$^3$ and S > 34.9)—estimated AW transport as 2.31 ± 0.29 Sv (**Table 1**). The authors also found large differences (from 0.53 ± 0.16 Sv to 3.39 ± 0.25 Sv) in transports amongst different transects conducted during their September survey, providing further evidence that the potential impact of eddies and meanders to these transports can be substantial.

The inflow of the Barents Sea water is thought to be the second potential forcing affecting the Laptev Sea transports; however, information about this inflow is very limited. Based on year-long mooring observations collected in the strait between Franz Joseph Land and Novaya Zemlya, the net volume transport of Barents Sea AW was estimated as ~1.9 Sv

(Loeng et al., 1997; **Table 1**). Unfortunately, we cannot verify the changes in these AW forcings along the EB slope to the east of St. Anna Trough, as there are no long-term observations of boundary current velocities that can be used for assessment of ACBC transports. Observations of currents at several sites near Novaya Zemlya and central Laptev Sea slopes in 2002-2012 have been derived from single moorings; these are not suitable for transport calculations, even if the period of

observations was long enough to provide insight for the long-term structure of the boundary current there (e.g., Pnyushkov et al., 2013; 2015). However, in the eastern part of the EB, at the junction of the Lomonosov Ridge with the continental slope, Woodgate et al. (2001) estimated water transports using year-long velocity mooring records from 1995-96. With some *a-priori* assumptions regarding the spatial structure of the ACBC between moorings, they estimated that the boundary current transports approximately 5 ± 1 Sv in the upper ~1200 m (i.e., within the layer spanned by velocity observations at moorings;

**Table 1**).

## 3 Observational data

### 3.1. Mooring observations and CTD surveys

In order to estimate transports in the eastern EB of the Arctic Ocean, we used observations collected in 2013–15 at six moorings distributed across the Laptev Sea continental slope (**Fig. 2**). These moorings, deployed along the 125°E meridian

in September 2013 as a part of the NABOS program, were designed to provide a detailed picture of structure and variability of the ACBC in this region (Polyakov et al., 2007; details of the observational program can be found at the project website - nabos.iarc.uaf.edu). The continental slope at the central Laptev Sea is gentle (bottom slope < 350 m/10 km), resulting in a lateral stretching of the ACBC. For an accurate description of the spatial ACBC pattern, the moorings were unevenly distributed across the slope, with shorter distances between the more shallow moorings. For example, the distance between

moorings $M1_1$ and $M1_2$ was approximately 11 km which is only 4 km larger than the first baroclinic Rossby radius of deformation estimated for the eastern EB (Nurser and Bacon, 2014). In the deep basin the distance increases to 170 km between $M1_4$ and $M1_5$ (**Fig. 2**). All moorings were successfully recovered in September 2015, providing a unique set of two-year-long records of velocity, temperature, and salinity observations. Summer mooring deployments and recoveries were accompanied by hydrographic surveys, using a *SBE-911plus* CTD profiler. These surveys provided detailed maps (with

distances between CTD stations varying from ~5 to 170 km) of the cross-slope structure of water properties at the Laptev Sea slope and adjacent regions.

Each mooring was equipped with instruments measuring horizontal velocities, temperatures, and conductivities (see mooring schematics in **Fig. 2** for details). Velocity observations were collected using McLane Moored Profilers (MMPs) and Acoustic Doppler Current Profilers (ADCPs). The MMPs operated between ~80 and 780-m depth every other day, with a

vertical resolution of approximately 0.25 m. Moving up and down along the mooring line at a mean rate of ~25 cm/s, it takes less than one hour for the MMP to measure the entire vertical profile within the above depth range. The relatively short

period of time required for the MMP to complete a single vertical profile suggests that each profile represents a snapshot of the hydrography and velocity at the mooring location. Each MMP was equipped with high-resolution (<0.01 cm/s) FSI (Falmouth Scientific Inc) acoustic current meter (ACM) sensor capable of measuring 2-D current velocities using the phase difference between upstream and downstream acoustic signals. Temperature, conductivity, and velocity sensors installed on-board the MMPs were calibrated by the MMP manufacturer before deployment in 2013.

All raw MMP data were processed using Woods Hole Oceanographic Institution (WHOI) software. During processing, the original measurements were averaged over 2-dbar pressure bins (the weights of the individual measurements were determined based on the pressure distribution within each bin). After averaging, the error of the bin-averaged velocities was approximately three times smaller when compared to the instrumental ACM error (~1% of the ACM reading ± 0.5 cm/s). In addition, raw MMP compass readings were corrected for the horizontal compass bias.

Seven upward-looking ADCPs also measured current velocities at the moorings. At all moorings except $M1_1$, the 300-kHz ADCPs were deployed at depths ranging from ~55 to 80 m to provide measurements of current velocities within the upper ocean layer. The uppermost ~6–10 m layer was left blank due to the interference of acoustic signals near the ocean surface. Additionally, 75-kHz ADCPs were installed at two moorings $M1_1$ and $M1_4$ that measured velocities in the layers of 5–245 m and 180-450 m, respectively. The ADCP records did not overlap with the MMP measurements (**Fig. 2**). ADCP measurements were averaged over two- (300-kHz ADCP) or five-meter (75-kHz ADCP) vertical cells, with one-hour time resolution.

In order to reduce the effect of measurement errors and to remove the tidal and inertial components of currents that dominate variability in the upper-ocean layer in this region (Pnyushkov and Polyakov, 2012), we averaged the hourly ADCP observations into daily records.

The magnetic inclinations determined from the International Geomagnetic Reference Field for the mooring positions (see www.ngdc.noaa.gov/IAGA/vmod/) were added to the raw readings of the magnetic compass of each instrument. The instrumental accuracy of the MMP magnetic compass is 2 °. The manufacturer's estimates for 75- and 300-kHz ADCP accuracies are 0.5 % of measured speed and 2 ° for current direction. However, due to the weak horizontal geomagnetic field strength in the EB, the individual compass error may substantially exceed the instrumental accuracy (errors may be as high as ~30 °; Thurnherr et al., 2017). These errors are individual for each instrument and cannot be quantified without concurrent (non-magnetic) measurements of current directions. Unfortunately, our moorings were not equipped with the instruments which could measure current directions that way, and, thus, we cannot provide more reliable estimates of the compass errors.

For each mooring, we merged the MMP and ADCP records into one data set that was further used to calculate transports. Replicating the temporal resolution of the filtered (low-passed) ADCP records, this dataset has a daily temporal resolution. Bi-daily MMP profiles were linearly interpolated in time to match the final temporal resolution. In the merged dataset, each

of the velocity profiles was accompanied by temperature and salinity profiles collected by the MMP and SBE-37 instruments (see **Fig. 2** for details of mooring instrumentation). Data gaps in temperature, salinity and velocity profiles in the layers between instruments were filled using vertical linear interpolation.

Due to time averaging, daily low-passed ADCP profiles contain less noise compared with snapshot bi-daily MMP velocity profiles. After averaging over the length of 2013–15 records, the largest (~0.3 cm/s) standard error of the mean velocity was observed at mooring $M1_3$. A sensitivity test was conducted to assess how the merging of data sets with different temporal resolutions and noise levels affects the estimates of net volume transports. In this test, for moorings equipped with MMPs (i.e., for $M1_2$, $M1_3$, $M1_5$, and $M1_6$ moorings) daily ADCP velocity profiles were replaced with snapshot measurements carried out by the ADCPs at times approximately matching the MMP snapshot measurements. Using this dataset, we estimated the net (within the upper 780-m layer) volume transport and found that the net eastward volume transport varied insignificantly ($< 5\%$).

We complement our analysis of mooring-based observations with an extensive dataset of temperature and salinity observations, collected from various expeditions in the eastern EB of the Arctic Ocean over the 2000–2015 period, to estimate the cross-slope structure of water masses over the EB slope (Sect. 4.1). This dataset was used, for example, in previous studies of long-term changes of the thermohaline state of the EB (Polyakov et al., 2008, 2012) and the structure of the ACBC (Pnyushkov et al., 2015). We note that the dataset was recently updated with CTD observations from two ship-based surveys, which accompanied the deployments and recoveries of moorings in 2013 and 2015 (see NABOS webpage at nabos.iarc.uaf.edu), and Ice Tethered Profiler (ITP) data (available at www.whoi.edu/itp). This dataset provides important information about temperature and salinity distributions across the continental slope of the Laptev Sea, and fills the gaps in the upper ocean layer, which was only partially covered by 2013–15 mooring observations. All oceanographic measurements have been made using CTD instruments with high accuracy for temperature (0.001 °C) and salinity (0.003).

### 3.2 Methods of analysis

Here, we provide details of the methods used in this study to calculate current persistence and geostrophic velocities as well as the volume, heat, and salt transports.

### 3.2.1 Persistence of currents

To quantify persistence (or stability) of current directions at the Laptev Sea slope, we calculated the number of days in the two-year records when daily currents fell into a specific directional bin. Each of these bins represents currents within a 12° angular sector. The width of these sectors is substantially wider than the reported instrumental accuracy of measurements for current directions, but may be comparable with those due to the weak horizontal magnetic field. In addition, we decomposed

velocity records for the mean and variable currents and calculated ellipses of standard deviations (SD) for the depth-averaged flow.

### 3.2.2 Geostrophic baroclinic currents

For estimates of the cross-section baroclinic geostrophic velocities (geostrophic shear velocities; $u_s$), we applied the thermal wind equations to the 2013–15 mean potential densities provided by mooring measurements. The same method was implemented in our previous studies of structure and variability of the ACBC at the Laptev Sea slope (see Pnyushkov et al., 2013 and 2015 for details). In these calculations, we, however, do not assume a level of no motion because during the 2013–15 period, the mean velocity profiles show no evident levels with zero velocities. This is likely due to a substantial barotropic flow at the Laptev Sea slope. Instead, we estimate the barotropic flow from the mooring records. Specifically, we calculate the depth average of eastward velocity for each mooring and average this barotropic velocity between each pair of moorings ($u_b$). We then add this to the baroclinic geostrophic velocity calculated from the thermal-wind relation, so that the total geostrophic currents were calculated as $u_g=u_b+u_s$.

### 3.2.3 Depth-integrated transports

Individually for each mooring, we vertically integrated the eastward velocities $u(z)$ to obtain water transport ($D_W$, m$^2$ s$^{-1}$), products of temperature anomaly $T(z)$-$T_{ref}$ and velocity ($D_H$, Wm$^{-1}$) to obtain heat transport, and products of salinity anomaly $S(z)$-$S_{ref}$ and velocity ($D_S$, kg m$^{-1}$ s$^{-1}$) to obtain salt transport (which we will call further *depth-integrated transports*).

$$D_W = \int_{z_l}^{z_{up}} u(z)dz \approx \sum_j 0.5(u_j + u_{j+1})(z_{j+1} - z_j), \ z_l \le z_j \le z_{up} \tag{1}$$

$$D_H = \int_{z_l}^{z_{up}} \rho c_p u(z)(T(z) - T_{ref})dz \approx \sum_j 0.5\rho c_p\left[u_j(T_j-T_{ref}) + u_{j+1}(T_{j+1}-T_{ref})\right](z_{j+1} - z_j), z_l \le z_j \le z_{up} \text{ and } \tag{2}$$

$$D_S = \int_{z_l}^{z_{up}} \rho u(z)(S(z) - S_{ref})dz \approx \sum_j 0.5\rho\left[u_j(S_j-S_{ref}) + u_{j+1}(S_{j+1}-S_{ref})\right](z_{j+1} - z_j), z_l \le z_j \le z_{up} \ , \tag{3}$$

where $z_l$ and $z_{up}$ are the lower and upper limits of integration, $c_p$ is the specific heat of sea water, and $\rho$ is the in-situ water density. $u_j$, $T_j$, and $S_j$ are the eastward velocity, temperature and salinity measured at the level $z_j$ , respectively. All these depth-integrated transports have simple physical meanings. For example, $D_w$ represents water transport within the specified depth range through a unit segment of mooring section. The integral of $D_w$ over the length of the mooring section provides the net volume transport. To calculate the products of temperature and salinity anomalies and velocities, available temperature and salinity profiles were interpolated to the levels of velocity measurements using linear interpolation. The transports of heat and salt were estimated using the freezing point $T_{ref} = $ -1.8 °C as the reference temperature and $S_{ref} = 0$ as the reference salinity. For calculations of the AW depth-integrated transports, $z_l$ and $z_{up}$ were determined as the shallowest and deepest levels with positive (T > 0 °C) water temperatures. If positive temperatures reached the deepest level of

observations we used the latter as the lower limit of integration. $D_W$, $D_H$, and $D_S$ at mooring M1$_4$ were estimated using extrapolated current velocities from 453 m (the deepest level of current observations) to the last CTD level (795 m) to extend data coverage within the AW layer. While extrapolating currents, we assume a barotropic vertical structure of the ACBC flow below 453 m so that all the extrapolated velocities were equal to the value measured at the deepest level with observations. The assumption of the barotropic vertical structure at mooring M1$_4$ is supported by velocity measurements from previous (2002–2011) years and by our estimates of geostrophic velocity in which the deviations from the uniform (barotropic) vertical profile below 450 m do not exceed 0.2 cm/s, or ~10% of the mean current at the mooring site.

In order to evaluate the limitations of this extrapolation in a quantitative way, we used velocity and temperature profiles consisting of depths ranging from 80–900 m at the M1$_4$ mooring site collected by the MMP instrument in 2004–05. Using these profiles, we calculated $D_H$ twice, once with original currents and another with extrapolated currents below 450 m, and compared the results. We found that the difference between the two $D_H$ estimates was small and did not exceed 5 %.

### 3.2.4 Along-slope transports

The property transports ($F_W$, $F_H$, and $F_s$) along the Laptev Sea section were estimated by horizontally integrating the depth-integrated transports over the length $L$ of the mooring transect (referred to as *transports* further in the text).

$$F_W = \int_{(L)} D_W \, dl \approx \sum_{i=1}^{5} 0.5(D_{W_i} + D_{W_{i+1}})\Delta l_{i,i+1}, \text{ [m}^3\text{s}^{-1} \text{ or Sv; 1 Sv} = 10^6 \text{ m}^3\text{/s]}, \tag{4}$$

$$F_H = \int_{(L)} D_H \, dl \approx \sum_{i=1}^{5} 0.5(D_{H_i} + D_{H_{i+1}})\Delta l_{i,i+1}, \text{ [W], and} \tag{5}$$

$$F_S = \int_{(L)} D_S \, dl \approx \sum_{i=1}^{5} 0.5(D_{S_i} + D_{S_{i+1}})\Delta l_{i,i+1}, \text{ [kg/s]}, \tag{6}$$

where $i$ indicates the mooring number, starting from the shallowest mooring M1$_1$; $\Delta l_{i,i+1}$ is the distance between the two neighboring moorings designated by $i$ and $i+1$.

The net transport was estimated for the entire area covered by mooring observations. For AW the integration was performed within the area spanned by this water mass. The AW transport uncertainty due to limited data coverage can be estimated if we extrapolate mooring velocities downward to the lower boundary of the AW (as determined by CTD profiles during mooring deployments and recoveries in 2013 and 2015). Applying the assumption of barotropic (depth-uniform) vertical structure of currents in this depth range, the inferred AW heat transport will increase by ~18 % (note that this number may change in case of using an alternate $T_{ref}$). This number provides a very rough estimate of the uncertainty for the AW transport due to incomplete data coverage.

Following the terminology widely accepted in oceanographic studies (e.g., Woodgate et al., (2006; 2010); Johns et al., (2011); Li et al., (2017)), we used the terms "heat" and "salt" transports to describe advective transports of water properties carried by the boundary current over the Laptev Sea section. However, we note that in the case of unclosed volume balance when the advected mass is not conserved, the $F_H$ and $F_S$ have meanings of temperature and salinity anomaly transports calculated relative to the reference values; only when the mass transport is balanced, these temperature and salinity anomaly transports have unambiguous physical interpretation as heat and salt transports. This limitation suggests that our estimates of heat and salt transports are valid only for the specific volume of water advected through the Laptev Sea section in 2013–15. We also note that the results of heat and salt transport calculations are sensitive to the choice of $T_{ref}$ and $S_{ref}$. Schauer and Beszczynska-Möller (2009) demonstrated that the uncertainty in the heat transports may be of the same order of magnitude as the signal itself. A similar dependence on $S_{ref}$ occurs in calculations of freshwater transports (see Tsubouchi et al., 2012; Carmack et al., 2016 for discussion). The requirement for mass conservation cannot be satisfied using observations from one cross-slope section; thus, our estimates for heat and salt transports are subject to uncertainty caused by an arbitrary choice of $T_{ref}$ and $S_{ref}$. However, our choice for $T_{ref}$ does have a clear physical meaning—the temperature below which seawater cannot exist as a liquid. Another reason for choosing the freezing temperature as $T_{ref}$ is that the heat content of the SML in winter is limited by this physical boundary. $S_{ref} = 0$ is also a reasonable choice, indicating the salt content in seawater; with that choice of $S_{ref}$ the calculated salt transport has unambiguous physical meaning even for a non-zero net volume transport. Despite limitations, our estimates have utility. In the case of using different $T_{ref}$ and $S_{ref}$, estimates for heat and salt transports can be recalculated using a simple linear relationship, originally suggested by Carmack et al. (2016) for freshwater fluxes. We note that the suggested relationship is also valid for heat and salt transports due to its linear nature relative to the reference values.

**3.2.5 Heat transport density**

We approach the problem of the sensitivity of heat transports to the choice of $T_{ref}$ for the case of unclosed mass balance by evaluating the heat transport density:

$$F_{HN} = \frac{\int_{(L)} D_H dz}{\int_{(L)} D_w dz} \approx \frac{\int_{z_1}^{z_2} \rho c_p u(z) T(z) dz}{\int_{z_1}^{z_2} u(z) dz} - \rho c_p T_{ref}, \qquad \text{[W s/m}^3 \text{ or TW/Sv]} \qquad (7)$$

$F_{HN}$ is the amount of heat transported by a unit of water transport, and thus is always related to a constant water transport. More generally, $F_{HN}$ quantifies the heat content transported by the mean current. As it follows from (7), a change in $T_{ref}$ shifts the mean of the entire $F_{HN}$ series, keeping the shape of the series intact. Therefore, for the assessment of temporal variability of the heat transports, using $F_{HN}$ is insensitive to the choice of $T_{ref}$ and, thus, is a more robust compared to an assessment made with $F_H$. In this study, we use both $F_H$ and $F_{HN}$ to quantify heat fluxes.

## 4 Water mass and flow structures over the Laptev Sea slope in 2013-15

In this section, we document the vertical thermohaline and dynamic structures of waters at the continental slope of the Laptev Sea, using temperature, salinity, and velocity profiles measured at the mooring array along the 125°E section in 2013–15, together with ship-based CTD measurements. Special attention is paid to the distribution of AW as the major contributor to water mass structure in the eastern EB.

### 4.1 Water mass structure over the Laptev Sea slope

In 2013–15, the AW occupied an intermediate layer between ~80 and ~1000 m (**Fig. 3**). On average, this water occupied ~80 % of the upper 780-m layer at the 125°E section, covered by the 2013–15 mooring measurements. Comparison of the 2013-15 mean temperatures against the PHC climatological temperatures for the early 1990s at the three deepest moorings (M1$_4$, M1$_5$, and M1$_6$) suggests a substantially warmer (>0.5 °C) AW core in recent years, in agreement with earlier findings (e.g., Polyakov et al., 2013; **Fig. 2, 3a**). The vertical position of AW boundaries in 2013–15 is also changed when compared with PHC climatology. The average depth of the AW upper boundary (defined by the 0 °C isotherm) was at ~125 m depth (shown as red dashed lines in **Fig. 3a, b**), which is ~50 m shallower than in the climatology. However, this boundary varied seasonally and spatially across the slope. For example, substantial seasonal displacement of the upper AW boundary was registered at mooring M1$_4$ (2700 m bottom depth), where it varied from ~85 m in January to ~170 m in early July (not shown). At mooring M1$_2$ (787 m), the upper AW boundary was significantly deeper, at ~200 m on average, but descending to ~350–400 m in middle summer. At the shallowest mooring M1$_1$ (250 m depth), AW was only present during a short period (<15 days) in winter of both years. This suggests that on average, the lateral AW boundary over the upper slope in 2013-15 was located between the M1$_1$ and M1$_2$ moorings.

At the three deep moorings M1$_4$, M1$_5$, and M1$_6$, the AW extends deeper than the last observational level (~780 m). Ship-based CTD profiles from 2002-15 along the 125°E line suggest the mean position of the lower AW boundary is ~800 m, which is close to the deepest observational level at NABOS moorings, but can vary significantly (down to ~1000 m) in some years. The relatively small differences between the deepest observational level at moorings and the lower AW boundary determined from the 2002–15 CTD casts at mooring sites suggest that our 2013–15 records include observations within 80 to 98 % of the AW layer, thus providing sufficient data coverage to describe water properties in this layer.

In 2013–15, the AW temperature core as determined by the in-situ temperature maximum (~220 m) was found at mooring M1$_5$ (**Fig. 3**). The AW core temperature averaged over the length of the mooring record was ~1.6 °C. AW temperature decreased gradually from the core toward both the shelf and deep basin. Temperature and salinity decrease much faster over the upper slope, between the M1$_3$ and M1$_2$ moorings. Higher spatial gradients of temperature and salinities between these moorings indicate the likely existence of a strong hydrographic front, separating waters from different origins, consistent with findings reported in Bauch et al. (2009). This front was evident, for instance, in the average distribution of 2013–15

potential density, where the average density gradients were as high as 0.26 kg/m$^3$ per 100 km (**Fig. 4**, lower panel; see Baumann et al., (2018) for further details).

Summarizing, we note that at least 80 % of the area covered by mooring observations in the upper 780-m layer of the Laptev Sea slope is occupied by the AW making it the dominant water mass there. However, the contribution from this water varies across the slope, from negligible at the upper slope (at the site of $M1_1$ mooring) to dominant at the deep part of the slope, where it occupies up to ~90 % (at $M1_4$ mooring site) of the water column between the surface and 780-m depth. The slope segments with dominant and weak AW contributions are separated by the hydrographic front found between $M1_2$ and $M1_3$.

### 4.2 Cross-slope pattern of velocities

The unique set of velocity records collected at the EB continental slope allows a comprehensive analysis of the spatial structure of the boundary current. We further assess the cross-slope pattern of the ACBC at the Laptev Sea slope using velocity profiles averaged over the record for all mooring sites (**Fig. 4**).

This analysis suggests that mean current directions agree well with the local topography of the Laptev Sea slope. For instance, the 2013–15 mean currents generally follow the isobaths with shallow-to-right direction; depths are from a two-minute global array of bottom topography (www.ngdc.noaa.gov/mgg/global/etopo2.html; **Fig. 5**). The agreement between mean current directions and topography is a known feature of a barotropic flow (Pedlosky, 1990), suggesting that mean (large-scale) circulation in the upper 780 m, including the AW layer in the eastern EB, is likely controlled by barotropic factors. The uniform vertical structure of the 2013–15 mean velocity profiles below 250 m at five deep moorings (except the shallow-water $M1_1$ mooring) supports our conclusion that the flow at the Laptev Sea slope is predominantly barotropic. The only exception is the cold halocline and the upper AW layers (above 200 m), where a substantial (up to 3 cm/s) baroclinic velocity maximum was found (**Fig. 4a**).

In agreement with the predominantly east-west orientation of isobaths at the central Laptev Sea slope, the zonal component of depth-averaged currents prevails over the meridional component for all moorings except $M1_6$ (**Fig. 4, 5**). As a result, we will refer to the zonal component of currents and transports as *along-slope*; the meridional component will be referred to as *cross-slope*. At $M1_6$, mean currents were directed approximately southward (**Fig. 5**). This southward direction for the mean currents at mooring $M1_6$ agrees with the pattern of AW circulation suggested by Rudels et al. (1994), who hypothesized the existence of cyclonic recirculation within the Amundsen Basin and the southward return branch of the AW along the Gakkel Ridge. This southward direction also agrees well with the distribution of bottom topography in which the flow goes along isobaths, similar to other moorings. The predominantly southward direction of the mean currents at this mooring suggests the boundary of the near-slope ACBC is somewhere between moorings $M1_5$ and $M1_6$.

The cross-slope distribution of mooring velocities shows a strong weakening of along-slope currents from the Laptev Sea shelf toward the deep ocean, from a maximum velocity of ~11 cm/s at the shallowest mooring ($M1_1$) to ~0.5 cm/s at mooring $M1_5$, and even to negative values at $M1_6$ (**Fig. 4**). The strongest (two-fold) reduction of velocities (from ~5 cm/s to 2 cm/s) at the shallow part of the Laptev Sea slope between the $M1_2$ and $M1_3$ moorings is accompanied by significant density changes associated with the hydrographic front (**Fig. 4d**).

Although mooring observations suggest a gradual decrease of eastward velocities from the shelf toward the deep ocean, the actual ACBC structure may have a more complex cross-slope pattern, which is not captured by the coarse mooring spacing. Simulations performed with the OCCAM (Ocean Circulation and Climate Advanced Modeling) numerical model suggest a multi-core ACBC structure in the EB (Aksenov et al., 2011). For example, the cross-slope distribution of simulated current velocities in the central Laptev Sea (along 126°E) demonstrates the existence of two separate velocity cores, evident in the eastward velocity component; one core is simulated above the lower part of the continental slope over the ~3000 m isobaths (near the $M1_4$ mooring site), while the second core is over the model continental shelf break at the ~400 m isobaths (see Fig. 8 in Aksenov et al., 2011). The velocity core over the deepest part of the continental slope was simulated within a depth range of ~200–400 m and is associated with the Fram Strait branch of the AW. The shallower branch transports Arctic halocline water, with negative water temperatures, along the EB slope within the ~100-200 m layer (Aksenov et al., 2011). We note here that Aksenov et al. (2011) suggested the existence of an additional BC core associated with Barents Sea Deep Water, which was formed by Barents Sea winter convection and located at ~900 m depth. However, available 2013–15 mooring records cannot confirm the suggested structure due to their limited spatial coverage.

The strongest velocity decrease at the hydrographic front suggests that the density distribution is firmly linked to the circulation in the intermediate layer in the eastern EB. For the area covered by deep mooring observations (at moorings $M1_4$, $M1_5$, and $M1_6$), the magnitudes of geostrophic currents contribute up to 80 % of the total current speed suggesting that the thermal-wind relationship is a good approximation in this region (**Fig. 6**). We complemented our estimates of the geostrophic currents over the Laptev Sea slope with their standard deviations as measures of confidence intervals (uncertainties) associated with the method utilized (see **Fig. 6**; dashed lines). In these calculations, we used daily temperature and salinity profiles at moorings to estimate geostrophic currents and their variability. Over the slope segment between the $M1_1$ and $M1_3$ moorings, the contribution of barotropic flow to the along-slope ACBC velocities is substantially larger when compared with the slope segments between deep moorings. This enhanced contribution is evident, for example, from the larger differences (exceeding one standard deviation) between the magnitude of baroclinic currents and measured velocities, reaching ~11 cm/s between the $M1_2$ and $M1_3$ moorings. At the sites of the shallow-water moorings $M1_1$ and $M1_2$ the vertical structure of the geostrophic velocities differs substantially from the vertical structure evident from the 2013–15 measured currents (**Fig. 6a**). The strongest discrepancies were found in the upper and bottom layers; likely suggesting that ageostrophic processes in the boundary layers play a role in the maintaining of vertical ACBC shape. Moreover, density gradients between moorings, which we used in our calculations of geostrophic currents, were estimated over the distances

exceeding the local baroclinic Rossby radius of deformation (~7 km; Nurser and Bacon, 2014), that may serve as an additional source of the disagreement between the geostrophic and measured currents because of synoptic-scale variability not resolved by our coarse mooring observations.

The distinct shape and relatively large (up to 10 cm/s; **Fig. 6a**) difference between the observed velocities and those derived
using geostrophic equations at shallow moorings $M1_1$, $M1_2$, and $M1_3$ suggest that the geostrophic balance in proximity to the density front may be violated.

## 4.3 Persistence of current velocities

The ACBC in the eastern part of the EB is subject to strong temporal variability, evident in both current speed and direction (Pnyushkov et al., 2015). Our analysis shows no exception, and strong variability of current directions is evident in the
velocity records from the mooring array collected in 2013-15.

The distributions of current speed and direction at the ~250-m level (close to the AW temperature core) demonstrate steady along-slope flow at the three shallowest moorings ($M1_1$, $M1_2$, and $M1_3$; **Fig. 7**). At these moorings, we found a relatively good match between the direction of the strongest currents (derived from all sectoral bins) and the directions of prevailing flow (black arrow; see moorings $M1_2$ and $M1_3$). In contrast, the persistence of current directions at the three deeper moorings
was significantly weaker—evident, for example, at $M1_5$, where there is essentially no prevailing current direction, and the mean and the strongest sectoral currents differ significantly (**Fig. 7**). Weak persistence of current direction over the deep part of the Laptev Sea slope is also confirmed by the distribution of standard deviation (SD) ellipses (**Fig. 5**). In contrast to the stretched shapes of these ellipses at $M1_1$ and $M1_2$ moorings, the ellipses at deep moorings have a circular shape, which suggests no prevailing direction of the flow.
Weaker persistence of currents at the three deep moorings (at $M1_4$, $M1_5$, and $M1_6$) is likely caused by weaker mean flow and stronger relative contributions from mesoscale variability. For example, observations carried out at the $M1_4$ mooring site in 2009–11 revealed multiple events of strong current rotation, when flow changes its direction by more than 90º over a short (from 4 to 15 days) period of time (Pnyushkov et al. 2018). These events potentially indicate the passage of eddies through the mooring site. Typical values of the inferred maximum rotation velocity in eddies registered at the $M1_4$ mooring site in
2009–11 are ~5 cm s$^{-1}$, although this value can exceed 15 cm s$^{-1}$, which is comparable, for example, to an eddy-induced current amplification found in the $M1_5$ mooring record in June–July 2014.

Topographically-trapped barotropic Rossby waves can also contribute to current variability. In the Arctic Ocean such waves have been observed both in the Canada basin (Timmermans et al., 2010) and over the Eurasian slope (e.g., Voinov and Zakharchuk 1999; Zakharchuk 2009). However, wavelet analysis of isopycnal displacements estimated using the MMP
record at mooring $M1_3$ located at the steep segment of the continental slope (not shown) does not show persistent spectral power (consecutive spectral power peaks) at typical periods of topographic Rossby waves (7-60 days; Zakharchuk 2009),

nor significant coherence between vertical isopycnal displacement and lateral velocity components. Such waves are therefore not likely to contribute strongly to the current variability found in our area of study.

## 5 Estimates of volume, heat, and salt transports across the 125°E section

Observations over the two-year (2013–15) period were used to estimate volume, heat, and salt transports in the eastern EB for the eastward (normal to the cross-section) direction of the flow.

### 5.1 Net volume transports across the 125°E section

Since the depth-integrated volume transport depends on both the current speed and water column height, shallower moorings may have weaker transport despite stronger currents. As an illustration, at the Laptev Sea slope, the largest monthly depth-integrated water transport ($55.6 \pm 1.5$ m$^2$/s; here, values after plus/minus signs indicate the standard error of the monthly mean) was found at mooring $M1_2$ and not at $M1_1$, which is caused by greater depth at this mooring location (**Fig. 8**; **Table 2**). Further north, depth-integrated water transports decrease to $20.1 \pm 0.6$ m$^2$/s at $M1_4$, and to $8.4 \pm 0.8$ m$^2$/s at $M1_5$. At $M1_6$, the record-mean along-slope water transport was small and negative ($-0.3 \pm 0.3$ m$^2$/s), resulting from the small westward component of the prevailing current there (see discussion in Sect. 4.2).

The net volume transport across the Laptev Sea slope was estimated using monthly averaged data and varies widely from ~$0.3 \pm 0.8$ to ~$9.9 \pm 0.8$ Sv (**Fig. 8b**). The largest (~$9.9 \pm 0.8$ Sv) net transport was observed during June-July 2014, and resulted from stronger depth-integrated transports at two deep moorings $M1_4$ and $M1_5$ (**Fig. 8a**, green and yellow lines). The mechanisms behind this transport amplification are not completely clear. In part, the strongest net volume transport in June–July 2014 was caused by a mesoscale eddy, which passed the $M1_5$ mooring site and significantly (up to 12 cm/s) increased eastward velocities in the entire layer spanned by mooring observations. The velocity increase was not evident at the neighboring moorings $M1_4$ and $M1_6$ due to the limited size of the eddy (comparable with the local Rossby radius of deformation). The contribution from shallow moorings in this transport enhancement was small, as suggested by a seasonal minimum in the depth-averaged transport at mooring $M1_2$ (**Fig. 8a**, blue line).

The contribution from different parts of the slope to the net volumetric water transport variability was evaluated by estimates of correlations between daily depth-integrated and net volume transports. These statistical estimates suggest that the strongest variability of along-slope transport occurs between the two deep moorings $M1_4$ and $M1_5$, likely due to the largest sectional area between them. For example, the time series for net volumetric transport and depth-averaged transports at moorings $M1_4$ and $M1_5$ are correlated with coefficients of 0.65 and 0.42 (both significant at the 95 % confidence level), respectively. Multivariate linear regression between the daily depth-integrated and net volumetric water transports suggests that the flow variability represented by these two moorings contributes up to 77 % of the net volume transport variability.

The high-frequency (from five to ten days) variations of depth-integrated transports is attributed to substantial mesoscale variability produced by eddies and ACBC meanders. Because of that, the correlations between daily series of depth-integrated transports at moorings are weak (not exceeding 0.2), except for the two closely located moorings $M1_1$ and $M1_2$, which are separated by a distance of ~11 km. For these two moorings, the correlation was as high as $0.6 \pm 0.1$. However, the

correlation between depth-integrated transports increases if the short-period (mostly incoherent) variability is removed. For example, after averaging over 30-day intervals (the longest period of averaging that enables confident statistics for correlation) the correlation coefficient for two deep moorings $M1_5$ and $M1_6$ (separated by more than 100 km) increases from $0.2 \pm 0.1$ to $0.4 \pm 0.1$. This suggests that variability at each mooring includes a substantial high-frequency component that is not resolved by our observations; however, relatively sparsely spaced moorings can successfully capture large-scale

dynamics that can be examined once incoherent mesoscale variability is removed.

## 5.2 AW transports

We estimated net volume transports for AW using temperature, salinity, and velocity measurements from the mooring array (**Fig. 8c**). Volume transport associated with the AW (the transport of waters with T > 0 °C) is ~$3.1 \pm 0.1$ Sv and contributes ~60 % to net volume transport. The relative contribution from AW transport is ~20 % smaller than expected from the

percentage of the area spanned by this water at the 125°E section. This difference is mostly due to the decrease in mean eastward velocities with depth (except the $M1_6$ mooring), so that in the AW layer velocities become smaller than those observed in the cold halocline and surface mixed layers. This weakening is evident, for example, at moorings $M1_2$ and $M1_3$ (**Fig. 4,** lower panel). For the $M1_6$ mooring at which larger eastward velocities were found in the deep layer (**Fig. 4a**), another factor at play is seasonal changes of the lower AW boundary. In 2013–15 this boundary (identified using the position

of a 0 °C isotherm) may shoal to 716 m so that the AW transport does not include the eastward transport in the layer below this depth. Additionally, there is substantial variability in the AW flow that includes periods of weak or even westward flow. For example, at mooring $M1_4$, depth-integrated volume transport in January 2015 was negative, indicating westward transport during this period.

AW transport at the Laptev Sea slope is highly correlated with net water transport in the upper 780 m with correlation of

$0.9 \pm 0.1$ (significant at the 95 % confidence level). As a result, the cross-slope pattern of the depth-integrated transports of the AW replicates the pattern of depth-integrated volume transport in the upper 780m layer shown in **Fig. 8a**. For example, the strongest depth-integrated AW transport was found at mooring $M1_2$—the shallowest mooring retaining a year-round presence of the AW—in January–February 2014, this transport exceeded 150 $m^2/s$, and decreased to almost zero during the summer months (not shown). Depth-integrated AW transports at deep moorings $M1_3$, $M1_4$, and $M1_5$ are smaller, and do not

exceed 70 $m^2/s$ due to a gradual decrease of the eastward component of the boundary current over the slope.

We note further that the largest volume transport (~1.2 Sv; **Fig. 8c**) was found at the 0–0.5 °C temperature range—at the boundary between the Fram Strait and the Barents Sea AW. This suggests that the estimates of AW transports are sensitive to the choice of temperature limits for this water. For example, a change in the temperature limit for the AW to 0.2 °C leads to a corresponding AW volume transport decrease of 0.5 Sv, constituting ~16 % of the mean AW volume transport (**Table 3**). However, for the accurate separation of AW branches at the Laptev Sea slope, more sophisticated analysis is required.

### 5.3 Salt and heat transports across the 125°E section

Estimated heat and salt transports across the 125°E section in the upper 780 m are shown in **Figs. 9, 10**. These estimates use along-slope (eastward) velocity components. The 2013–15 salt transport in the upper 780-m layer is estimated as $172 \pm 6$ Mkg/s. In the AW layer, salt transport is $112 \pm 4$ Mkg/s constituting ~65 % of the net salt transport. The use of higher value of $S_{ref}$ increases the relative contribution of the AW layer to the net salt transport. For example, for $S_{ref} = 31.7$ (the minimum salinity measured at moorings) we found that this contribution increases to 82%. The major statistics for these transports are summarized in **Tables 3, 4**. The 2013–15 heat transport carried by the AW was $32.7 \pm 1.3$ TW, constituting ~71 % of the net heat transport in the entire layer spanned by mooring instruments ($46.0 \pm 1.7$ TW); ~11 % higher than the relative contribution from AW to the net volume transport. This high percentage suggests a substantial portion of heat and salt in the eastern EB is carried by the AW. In the AW layer, heat transport density is larger than in the entire layer spanned by 2013–15 observations (**Fig. 9c**). The two-year mean heat transport density for the AW was $6.8 \pm 0.3$ TW/Sv, which is ~26 % larger than for the net heat transport density of $5.0 \pm 0.2$ TW/Sv, which also includes fresh and cold waters of the SML and CHL.

The time series of net salt and heat transports in the AW layer are highly correlated with AW volume transports ($R > 0.9$, significant at the 95 % confidence level). Similarly high correlations were also found for the series of depth-integrated heat and salt transports at all moorings. Since the heat and salt transports are influenced by two major factors—the speed of currents and temperature/salinity anomalies (the AW layer thickness changes by 5% only and has a minor role in changes of the AW property transports)—these high correlations suggest the variability of currents plays the most significant role in governing heat and salt transports changes in this part of the EB. Based on the correlations between transports, we conclude that the currents are responsible for ~90 % of the variability of heat and salt transports at the scales resolved by our records (in this estimate we used variance as a qualitative measure of variability). The remaining ~10 % is controlled by temperature and salinity anomalies together with temporal variability of the AW layer thickness. The level of correlation between the volume and heat transports is high for the current choice of $T_{ref}$, but remains high for alternate choices (e.g., $R>0.8$ for $T_{ref}=0$ °C). The multivariate linear regression between daily depth-integrated heat transports and mean temperatures/salinities within the AW layer used as the predictors and $F_H$ suggests approximately the same (>90 %) contribution from currents (estimated as the sum of the derived coefficients of determination for $D_W$) to the net heat transport.

Summarizing, we conclude that AW contributes substantially (from 60 to 71 %) to the net water, heat, and salt transports along the Laptev Sea slope in the upper 780 m. However, the estimates of heat and salt transports and their relative contributions are subject to uncertainties related to the unclosed mass balance. Spatiotemporal variability which is not resolved by available mooring records adds uncertainty to our estimates of transports.

## 5.4 Temporal variability of transports

### 5.4.1 Year-to-year changes

In 2013–15 the net volume, heat, and salt transports across the Laptev Sea slope experienced strong temporal changes. During this period, the net volume transport decreased from $5.8 \pm 0.2$ Sv in 2013–14 to $4.4\pm0.2$ Sv in 2014–15, and the AW volume transport reduced correspondingly from $3.7 \pm 0.2$ Sv to $2.6\pm0.1$ Sv (**Fig. 8**). Because of the limited length of the
available records, these estimates may not be representative of transports across longer time scales. Declines were also found in the net and AW heat transports (**Fig. 9b**), likely replicating the decreased water transports across the Laptev Sea slope. To provide more robust estimates of year-to-year changes in heat transports in the case of decreased and (in addition) highly variable water transports, we have computed time series of heat transport density. Since both the heat transport and heat transport density quantify heat fluxes, their time series are highly correlated (the coefficients of linear correlation are
$0.8 \pm 0.1$ and $0.7 \pm 0.1$ for the heat transport density for the upper 780 m and in the AW layer, respectively). Taking into account that the heat transport density is normalized using the volume transport, this high correlation suggests that the temperature regime in the AW layer, including heat fluxes, is strongly coherent with the volume of AW advected through the Laptev Sea slope. Following the heat transport, the heat transport density demonstrates a slight decrease from $7.8 \pm 0.4$ TW/Sv in 2013–14 to $6.3 \pm 0.4$ TW/Sv in 2014–15, though the estimated decrease rate is statistically insignificant (using the
Student's t-test) making our conclusion about the decline of heat transport across the Laptev Sea slope more speculative (**Fig. 9c**).

In order to better assess year-to-year transport changes, all depth-integrated and volumetric transports were estimated individually for the first (September 2013–August 2014) and second (September 2014–September 2015) years. Comparison of annual mean estimates for depth-integrated and net transports of volume, heat, and salt at moorings confirms substantial
variability of these transports. For example, at mooring $M1_4$, where the strongest year-to-year changes in the depth-integrated volume transport occurred, this transport decreases by an order of magnitude, from $15.0 \pm 1.0$ in 2013–14 to $1.3 \pm 1.0$ m$^2$/s in 2014–15, resulting in a corresponding decrease in net volume transport (**Table 2, 3**). This strong decrease was due to the change to the north-west of the predominant direction of the flow evident at the $M1_4$ mooring site in February–March 2015, alongside the substantial (up to 15 cm/s) enhancement of the daily current speed within the 180–400
m layer. In this period, the flow demonstrated pronounced baroclinic vertical structure (not shown), at which the largest velocities were found at ~450 m level (at the last level with velocity observations). These changes in the volume transport

resulted in the decrease of heat and salt transports. For instance, net heat (salt) transports decreased from $55.3 \pm 2.8$ TW ($195 \pm 8$ Mkg/s) in 2013-14 to $38.8 \pm 2.1$ TW ($147 \pm 7$ Mkg/s) in 2014-15.

## 5.4.2 Seasonal and monthly variability

In addition to significant year-to-year changes, volume transports demonstrated substantial seasonal variations. We determined these changes by employing Morlet wavelet power spectra to depth-integrated AW transports at moorings (**Fig. 11**; Grinsted et al., 2004). The focus on the AW layer was due to the seasonal mode of variability being dominant in this layer at the continental slope of the eastern EB in 2008–2011; the seasonal mode described up to 70 % of variability in the AW temperatures and ACBC velocities (Pnyushkov et al., 2015).

However, 2013–15 mooring records show a clear seasonal signal indicated as maximum of wavelet power spectra at periods from ten to twelve months in the depth-integrated AW transport only at the two shallow moorings $M1_2$ and $M1_3$ (**Fig. 11**). The relative contribution of seasonal changes to depth-integrated transport variability varies from ~30% at the $M1_2$ mooring site to ~11% at $M1_4$ mooring (estimated from the ratio of wavelet power at the seasonal scales to the total wavelet power; **Fig. 11**). The seasonality in AW transport at $M1_2$ and $M1_3$ mooring may be connected to significant seasonality in temperatures and velocities also observed over the upper slope north-east of Svalbard, with maximum values in late fall to early winter (Renner et al., 2018).

Substantial power for all mooring records is evident in the wavelet spectra of the AW transports at periods from ~30 to ~250 days (**Fig. 12**). Note that strong variability at the same scales was also found in current speed records at NABOS mooring M4 in 2004–05 located at the north-east slope of Svalbard (~30°E; unpublished results) and at A-TWAIN (Long-term variability and trends in the Atlantic Water inflow region) moorings in 2012–13 (Renner et al., 2018). Moreover, strong variability at scales from one to several months was found in wavelet power of baroclinic velocities calculated using temperature and salinity from the two deep moorings $M1_4$ and $M1_5$ (**Fig. 12**). These moorings were chosen because they provide the highest correlations with the net AW transport across the Laptev Sea slope, and thus describe the dominant part of AW transport variability. Both wavelet power spectra have similarities at time scales from 50 to 100 days, with similar statistically significant peaks of wavelet power for both baroclinic currents and AW transports in 2013–14 and, to a lesser extent, in 2014–15 (**Fig. 12**). This suggests that variability of the cross-slope density gradients is coherent with the variability of transports at these scales in this part of the EB. We have not found any statistical relationship between the AW depth-integrated transports and the atmospheric forcing. For example, correlations between the ERA Interim (Simmons et al., 2006) daily wind velocities and wind vorticity with ocean transports for all moorings were low ($R < 0.2$), and were accompanied by different wavelet patterns (not shown).

### 5.4.3 Concluding remarks on transports

New observations have been used to demonstrate that water, heat, and salt transports across the Laptev Sea slope exhibited strong changes in 2013–15—as evident, for example, from the strong year-to-year changes in net volume transport (**Fig. 8**). These changes were most pronounced at deep moorings, where the year-to-year difference in depth-integrated transports can be as large as the record-long means. The year-to-year changes in volume transports were accompanied by substantial monthly changes, evident also in cross-slope baroclinic geostrophic velocities, suggesting that density-driven currents are the likely contributor to the variability of transports at monthly temporal scales in this part of the EB.

## 6 Discussion and conclusions

### 6.1 Along-slope transports

Mooring observations suggest that in the upper 780m layer, the ACBC advects ~5.1 ± 0.1 Sv of water, predominantly in the eastward direction, and ~3.1 ± 0.1 Sv (or ~70 %) of this volumetric transport is associated with AW (T>0.0 °C) – the dominant water mass at the Laptev Sea slope. The direction of 2013–15 mean water transport agrees well with local topography. Up to 70% of the net volume transport in the eastern EB in the upper 780-m layer is carried by the AW.

These estimates of the net volume transport at the Laptev Sea slope agree relatively well (within the range of uncertainties for the means) with the 5 ± 1 Sv of advective transport at the junction of Lomonosov Ridge and the Siberian shelves estimated by Woodgate et al. (2001) in the upper ~1200-m layer for 1995–96. This estimate is only 0.1 Sv smaller than our estimate for the 2013–15 net water transport across the 125°E section. Approximately the same volume transport over a ~30% broader depth range at the Lomonosov Ridge suggests that the flow in recent years was enhanced when compared with the mid-1990s. The same difference (~0.1 Sv) was found between the AW transport at the Laptev Sea slope and that reported in Fram Strait, where mooring-based observations collected in 1997–2010 suggest a mean northward (into the Arctic Ocean) transport of AW (waters warmer than 2 °C) of 3.0 ± 0.2 Sv (Beszczynska-Möller et al., 2012).

A comparison of the 2013–15 mean AW transport at the Laptev Sea slope with estimates available for the northeast slope of Svalbard suggests large (but within one standard deviation range) differences in these transports. According to Våge et al. (2016), the snapshot AW transport in the area to the north of Svalbard was 1.6 ± 0.3 Sv in September 2012, differing by ~1.5 Sv (~50 %) from our record-mean AW transport across the 125°E section, but well within the range of synoptic-scale spikes in our record (**Fig. 8**). A smaller difference (~0.8 Sv) was found for the estimate reported by Pérez-Hernández et al. (2017), who reported along-slope AW transport of 2.31 ± 0.29 Sv, based on synoptic-scale cross-slope CTD sections between 21°E and 33°E. The plausible reason for those differences is different time scales of observations used for transport estimates (i.e., synoptic scale at Svalbard and annual scale at the Laptev Sea slopes). Moreover, a seasonality of ocean

currents to the north of Svalbard with amplified AW current speed in winter suggests the annual AW transport is higher when compared with the reported early-fall estimates (Randelhoff et al., 2015; Renner et al. 2018). In addition, all reported AW transports estimated in the proximity to Svalbard were based on geostrophic calculations, and thus may underestimate the barotropic component of the flow, from which contributions to the net water transport may be significant.

The mean heat transport carried by AW along the Laptev Sea slope was quantified as high as 32.7 ± 1.3 TW. At the Laptev Sea slope, AW heat transport constitutes ~71 % of the net heat transport (46.0 ± 1.7 TW), confirming a dominant role for AW heat in the thermal balance of the intermediate ocean layer in the EB. Moreover, since velocity and CTD observations used for calculating heat transports at moorings do not always cover the entire layer spanned by the AW (see Sect. 2 for details), the actual AW heat transport may be even higher.

Estimates of transports using the sparsely-spaced moorings may suffer from cross-slope ACBC meandering. In order to estimate the potential impact of meandering on volume transports, we used simulations performed with a state-of-the-art Estimation of the Circulation and Climate of the Ocean (ECCO) numerical model (Nguyen et al., 2017). This global model has a 4-km spatial resolution, which is capable of capturing possible cross-slope migration of the ACBC velocity core. The model was successful in demonstrating a synergy between observations carried out from drifting platforms in the Arctic
Ocean and coupled ocean-sea ice modeling, and improving the circulation pattern in regions with sparse data coverage, such as the eastern Arctic and the seasonal ice zones (for details, see Nguyen et al., 2017). For the eastern EB, the model reproduces a velocity pattern with two cores, the deep velocity core was over 1200–1700 m bottom depths—at the slope segment between the $M1_2$ and $M1_3$ moorings; the shallow velocity core was simulated over 350–400 m bottom depths. The simulated ACBC meandering at the Laptev Sea slope is estimated by calculating the standard deviation of the position of the
deep ACBC velocity core. We found that the estimated standard deviation is small, less than 8 km. We also compared 2004–15 simulated mean volume transports estimated within the upper ~800-m layer using all model grid points along the mooring section with those made using a few grid points corresponding to the sites of NABOS moorings only. This comparison showed a 6% difference which suggests that transport that is unresolved by moorings due to cross-slope meandering of the velocity core is small.

**6.2 Potential mechanisms of transport variability**

The complex nature of the ACBC leads to a diversity of hypotheses for the physical mechanisms that control the transports in the eastern EB. For example, at longer (inter-annual) time scales, the variability may be dominated by barotropic forces (e.g., advection of potential vorticity with the Barents Sea branch of the AW) as suggested by good agreement between the mean current in the upper 780 m layer with local topography, a feature of the barotropic boundary current aligned to the
continental slope (**Fig. 5**). AW transport during 2013–15 at the Laptev Sea slope agrees well with that reported for Fram Strait moorings (Schauer et al., 2004, 2008; Beszczynska-Möller et al., 2012), likely suggesting a causal linkage between

water transports at both sites at annual scales. However, due to the limited length of mooring records at the Laptev Sea section, we cannot quantify this relationship statistically.

At shorter (monthly to seasonal) time scales, variability is likely linked to changes in water density gradients, as suggested by the similarity of the wavelet power spectra of geostrophic velocities and depth-averaged AW transports at time scales from 50 to 100 days (**Fig. 12**).

Atmospheric circulation is also an important factor driving oceanic circulation at various scales. For example, Nøst and Isachsen (2007) suggested the influence of wind curl in the Nordic Sea on the ACBC circulation in the Arctic Ocean. For the eastern EB, wind pattern has an important or even dominant influence on the circulation and freshwater redistribution in the upper layer (Morison et al., 2012). However, we did not find a statistical relationship between ACBC transports along the Laptev Sea slope and local winds. A weak relationship between local winds and currents at the Laptev Sea slope was also reported earlier for the central Laptev Sea slope (~125°E) by Pnyushkov et al. (2015), who noted about weakly correlated winds and ACBC velocities within the AW layer and in the halocline. However, the atmospheric impact on volume transports may be non-local. For instance, based on a year-long current record in the St. Anna Trough, Kirillov et al. (2012) suggested that the changes in atmospheric circulation regime in the northern parts of the Barents and Kara seas influences the intensity of Barents Sea AW exchange with the deep EB, likely impacting along-slope transports at the Laptev Sea slope. To quantify non-local wind impact, we calculated lagged correlations (with a maximal lag of 36 months–this time is presumably enough to advect the signal from Fram Strait to the Laptev Sea slope) between weekly ERA Interim winds (zonal and meridional wind components and wind speed) and the net volume transports at the Laptev Sea slope. The wind components were taken for all available reanalysis nodes located in the EB from Fram Strait to the central Laptev Sea. The highest correlations with the wind at the sites of reanalysis nodes were low ($R=0.27\pm0.15$), but statistically significant, with a spotty spatial pattern of the time lags at which those correlations were found. These low correlations likely suggest that the non-local atmospheric impact on property transports in the eastern EB cannot be identified robustly at temporal scales resolved by our relatively short (two-year long) records.

### 6.3 Limitations of the analysis

Estimates for the heat, salt, and volume transports reported in this study are subject to some caveats. For example, the sparse spacing between moorings, which varies from ten to more than a hundred kilometers, does not allow for the interpretation of some mesoscale features, such as the eddies and meanders found in single mooring records and depth-averaged transports. Their accurate description requires knowledge of the horizontal pattern of ACBC flow; thus, higher spatial resolution would provide more accurate transport estimates (Våge et al., 2016; Pérez-Hernández et al., 2017). The mesoscale variability evident in our mooring records masks the low-frequency signal, reducing coherence between daily depth-averaged transports; however, this coherence increases when monthly averaging is applied.

The estimates of AW transports are also sensitive to the temperature and salinity ranges used for the identification of this water. At the central Laptev Sea slope, AW comprises water from the Fram Strait and Barents Sea AW branches and the product of their mixing. However, we still do not have reliable criteria for separating waters of these two branches, taking into account their complex transformations and interactions on the way from upstream locations to the Laptev Sea. At the same time, a good match between volume transports at the central Laptev Sea slope (3.1 Sv) and in Fram Strait (~3.0 Sv) indicates that the selected definition of AW (T > 0 °C) is reasonable.

The limited length of 2013–15 NABOS mooring records restricts our ability to understand the important features of transport variability at temporal scales not resolved by the records. An evident decrease in transports in 2014–15 in comparison with 2013–14 is robust, but cannot be considered representative for longer time scales. Estimates of substantial year-to-year changes of net and AW transports across the slope in the eastern EB are also trustworthy and agree well with dominant variability at the same scales reported for Fram Strait transports (Schauer et al., 2008).

### 6.4 Concluding remarks

Despite the limitations for analysis, we argue our study is an important step toward a better understanding of the dynamics of the Arctic Ocean, as it provides the first observational estimates for heat, salt, and volume transports along the Laptev Sea slope. All these transports represent vitally important information for the validation of state-of-the-art numerical models.

Two years of mooring observations suggest that the AW dominates in the ACBC heat, salt, and water transports over the Laptev Sea slope; these transports are likely controlled by a host of factors acting at different time scales. We found that the low-frequency (year-to-year) changes are the most pronounced modes of variability for the transports at the Laptev Sea slope in recent years—the time when significant changes of the thermohaline state and circulation occur in the eastern EB (e.g., Polyakov et al., 2013; 2017; Pnyushkov et al., 2015). An increased role for Atlantic inflows in those changes suggests that the eastern EB is now in a transition to a new thermohaline state which is similar to that in the western part of the EB and in the Nordic Seas (see Polykov et al., 2017 for discussion). However, the substantial year-to-year changes in AW transports found at the slope of the Laptev Sea in 2013–15 likely suggest that robust documenting of longer-term tendencies of these transitions require further observations.

### Acknowledgments

The authors thank the Editor, Drs. M. Luneva and B. Rabe for assistance in evaluating this paper. This study was supported by NSF grant #1708427 (A. Pnyushkov, I. Polyakov). The mooring- and ship-based oceanographic observations in the eastern EB were conducted within the framework of the NABOS project, with support from NSF (grants AON-1203473, AON-1338948, and AON-1203146). V. Ivanov acknowledges funding from the Ministry of Education and Science of the

Russian Federation (project RFMEFI61617X0076). G. Alekseev was supported by the RFBR grant 18-05-60107. A. Sundfjord was supported by the Fram Centre 'Arctic Ocean' A-TWAIN project.

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

**Table 1. Estimates of volume transport in the Eurasian Basin of the Arctic Ocean.**

| Location | Period | Water properties | AW volume transport, Sv | Source |
|---|---|---|---|---|
| Fram Strait | 1997–2019 | T>2 ℃ | 3.0 ± 0.2 | Beszczynska-Möller et al., 2012 |
| North of Spitsbergen (near 30ºE ) | Sep. 2012 | $27.7 < \sigma < 27.97$ kg/m$^3$ and T > 2 ℃ | 1.6 ± 0.3 | Våge et al., 2016 |
| North of Spitsbergen (between 21ºE and 33ºE) | Sep. 2013 | $27.7 < \sigma < 27.97$ kg/m$^3$ and S > 34.9 | 2.31±0.29 | Pérez-Hernández et al., 2017 |
| The strait between Franz Joseph Land and Novaya Zemlya | Sept. 1991– Sept. 1992 | Barents Sea AW | ~1.9 | Loeng et al., 1997 |
| Eastern EB (78°31'N and 133°58'E) | 1995–96 | net transport in the upper 1200-m layer | 5 ± 1 | Woodgate et al., 2001 |

**Table 2.** Depth-integrated water transports (along-slope, $m^2/s$) in the 0–780 m layer in 2013–15 at six moorings at the Laptev Sea slope.

| Period | $M1_1$ | $M1_2$ | $M1_3$ | $M1_4$ | $M1_5$ | $M1_6$ |
|---|---|---|---|---|---|---|
| Sep 2013–Aug 2014 | 15.4±0.9 | 52.9±2.2 | 19.4±0.9 | 15.0±1.0 | 5.4±1.1 | -2.9±0.4 |
| Sep 2014–Sep 2015 | 22.4±1.0 | 58.6±2.1 | 20.9±0.9 | 1.3±1.0 | 3.4±0.7 | 2.6±0.4 |
| Total (Sep 2013–Sep 2015) | 18.7±0.7 | 55.6±1.5 | 20.1±0.6 | 8.4±0.8 | 4.5±0.7 | -0.3±0.3 |

**Table 3. Net heat, salt, and volume transports across the 125ºE section in the upper 780-m layer in 2013–15.**

| Period | Volume transport, Sv | Heat transport, TW | Salt transport, $\times 10^6$ kg/s |
|---|---|---|---|
| Sep 2013–Aug 2014 | 5.8±0.2 | 55.3±2.8 | 195±8 |
| Sep 2014–Sep 2015 | 4.4±0.2 | 38.8±2.1 | 147±7 |
| Total (Sep 2013–Sep 2015) | 5.1±0.1 | 46.0±1.7 | 172±6 |

**Table 4. Atlantic Water heat, salt, and volume transports across the 125ºE section in 2013–15.**

| Period | Volume transport, Sv | Heat transport, TW | Salt transport, $\times 10^6$ kg/s |
|---|---|---|---|
| Sep 2013–Aug 2014 | 3.7±0.2 | 38.6±2.0 | 131±7 |
| Sep 2014–Sep 2015 | 2.6±0.1 | 26.4±1.6 | 92±5 |
| Total (Sep 2013–Sep 2015) | 3.1±0.1 | 32.7±1.3 | 112±4 |

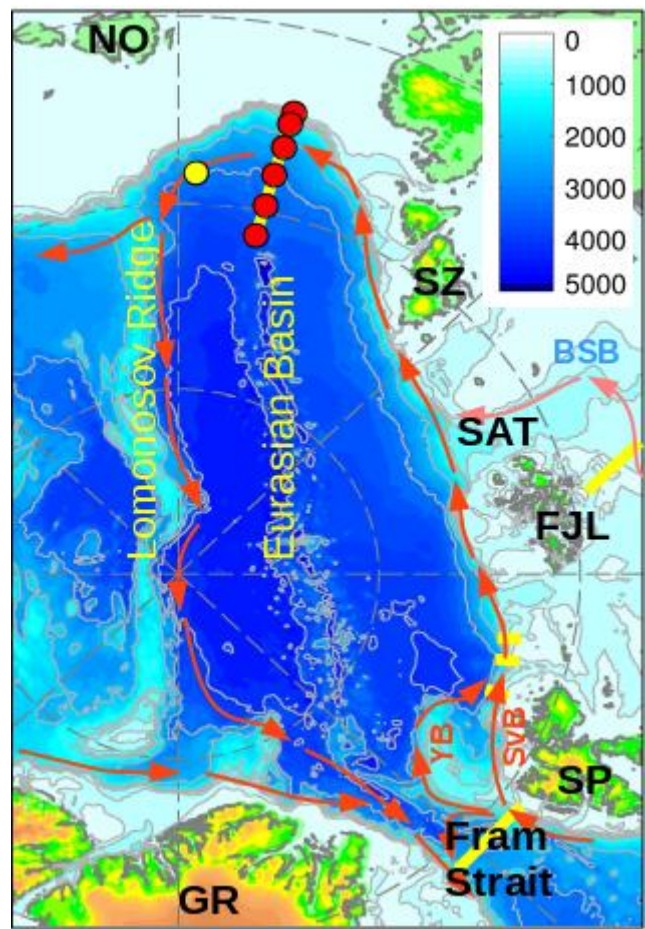

**Figure 1: Map showing location of moorings (red circles) over the continental slope of the Laptev Sea in the Eurasian Basin of the Arctic Ocean in 2013–2015. Greenland (GR), Spitsbergen (SP), Franz Joseph Land (FJL), St. Anna Trough (SAT), Severnaya Zemlya (SZ), and Novosibirskiye Islands (NO) are indicated. YB, SvB, and BSB indicate Yermak, Svalbard, and Barents Sea Atlantic Water (AW) branches, respectively. Red arrows show a schematic pattern of AW circulation in the Eurasian Basin. Bottom depth in meters is shown by color. Different sections and moorings mentioned in the paper are indicated with yellow lines and circles.**

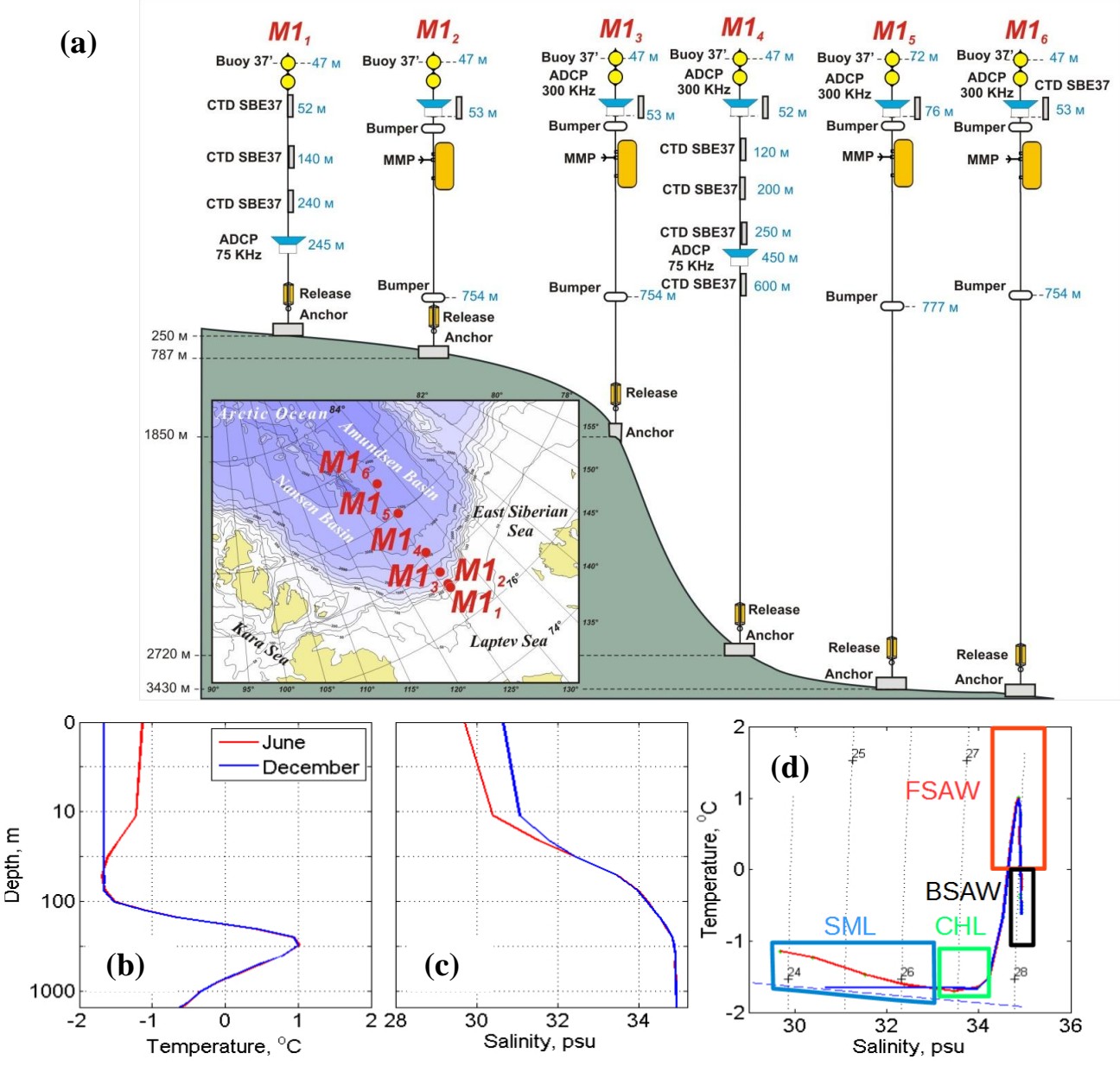

**Figure 2: (a) Schematics and locations (insert) of six moorings deployed as a cross-slope array in the eastern Eurasian Basin along 125ºE in 2013–15. Note that the distances between the moorings shown in these schematics do not reflect real distances (as shown in insert). (b, c) Climatological potential temperature (θ) and salinity (S) profiles, and (d) θ-S diagram for the central Laptev Sea slope area (77–79ºN; 125–126ºE). In (d), gray dotted lines show isopycnals, and blue dashed line shows freezing temperature. Vertical distribution of the climatological potential density resembles salinity distribution shown in (c). Color rectangles emphasize different water masses occupying the Laptev Sea slope.**

SML, CHL, FSAW, and BSAW in the T-S plot indicate the Surface Mixed Layer, the Cold Halocline Layer, the Fram Strait Atlantic Water, and the Barents Sea Atlantic Water, respectively

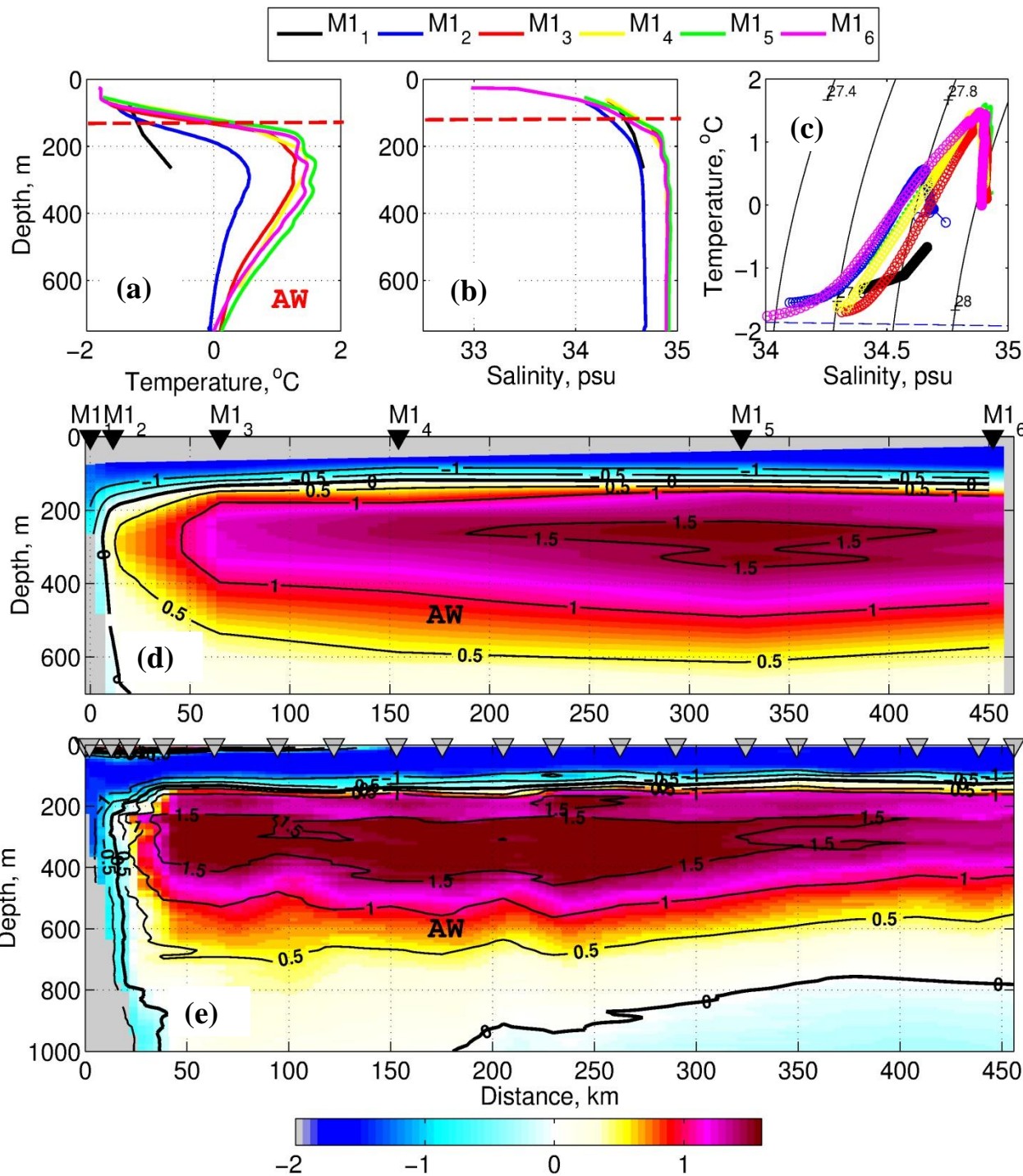

**Figure 3.** 2013–15 mean (a) temperature (T) and (b) salinity (S) profiles, and (c) T-S diagrams from mooring array deployed at the Laptev Sea slope. Red dashed lines show position of the AW upper boundary. Mean temperature

distribution along the 125ºE section from (d) the 2013–15 mooring observations, and (e) 2013 summer CTD section. Black thick lines in (d) and (e) show the boundaries of the AW layer. Gray triangles in (e) show positions of CTD stations in 2013.

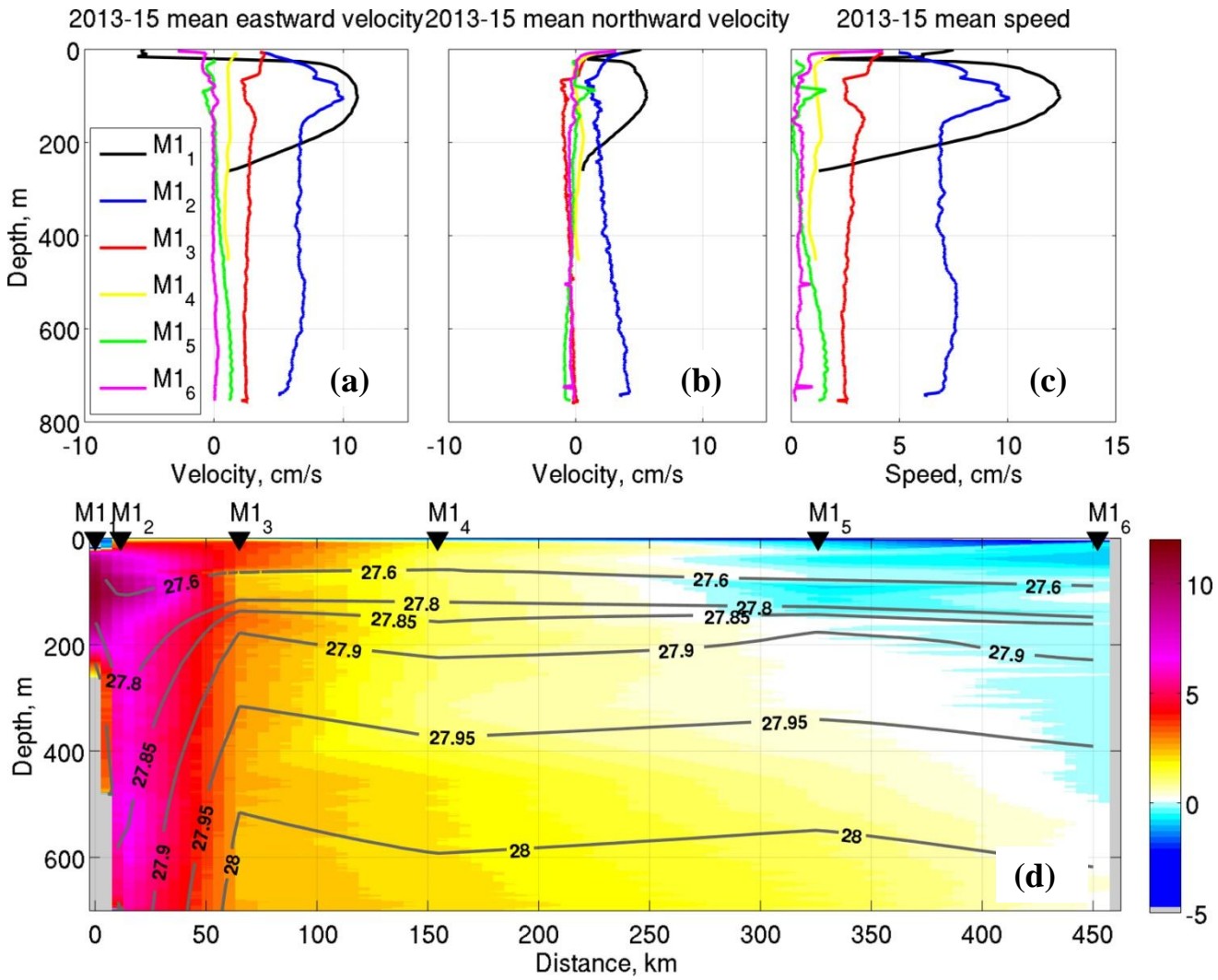

**Figure 4: (a) Eastward and (b) northward velocities, and (c) current speed at six moorings deployed at the Laptev Sea slope in 2013–15; (d) distribution of the mean eastward velocity along the 125ºE section. Grey contours show mean potential density calculated using 2013–15 temperatures and salinities observations at moorings.**

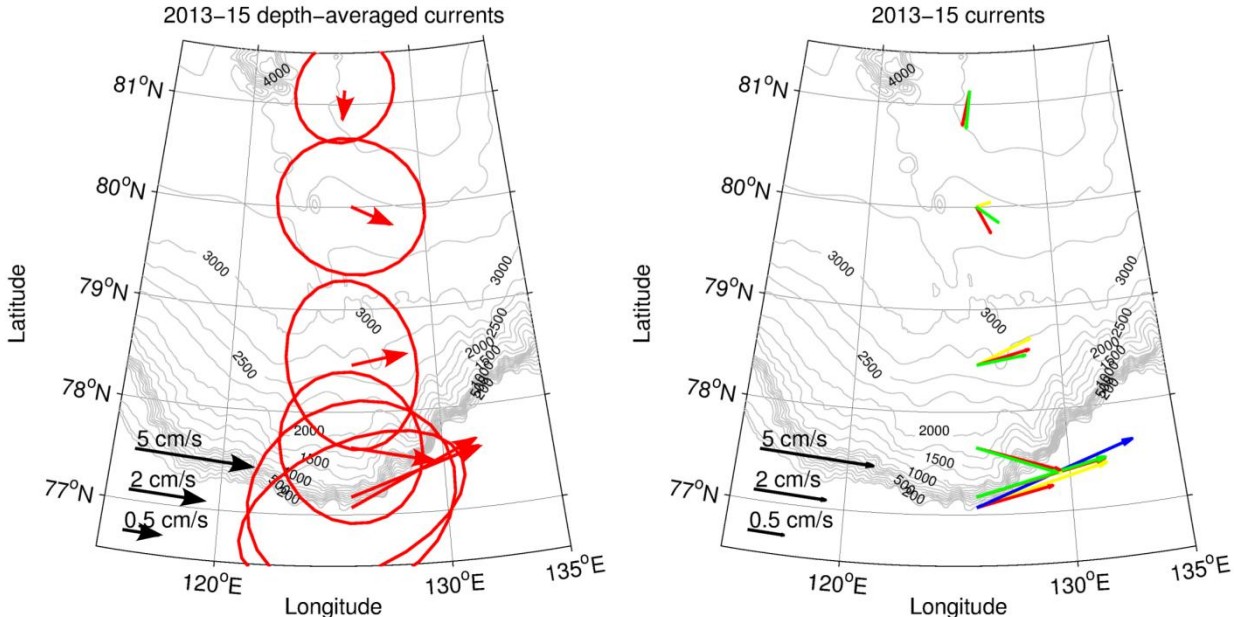

**Figure 5: (Right) 2013–15 mean current vectors at 150 (blue), 200 (yellow), 250 (red), and 300 (green) meters, and**
**(left) their depth-averaged (0–780 m) means. Grey contours show isobaths. Red ellipses with the centers at mooring**
**sites represent ellipses of standard deviations of the depth-averaged currents. A non-linear scale is used to show**
**vectors and ellipses.**

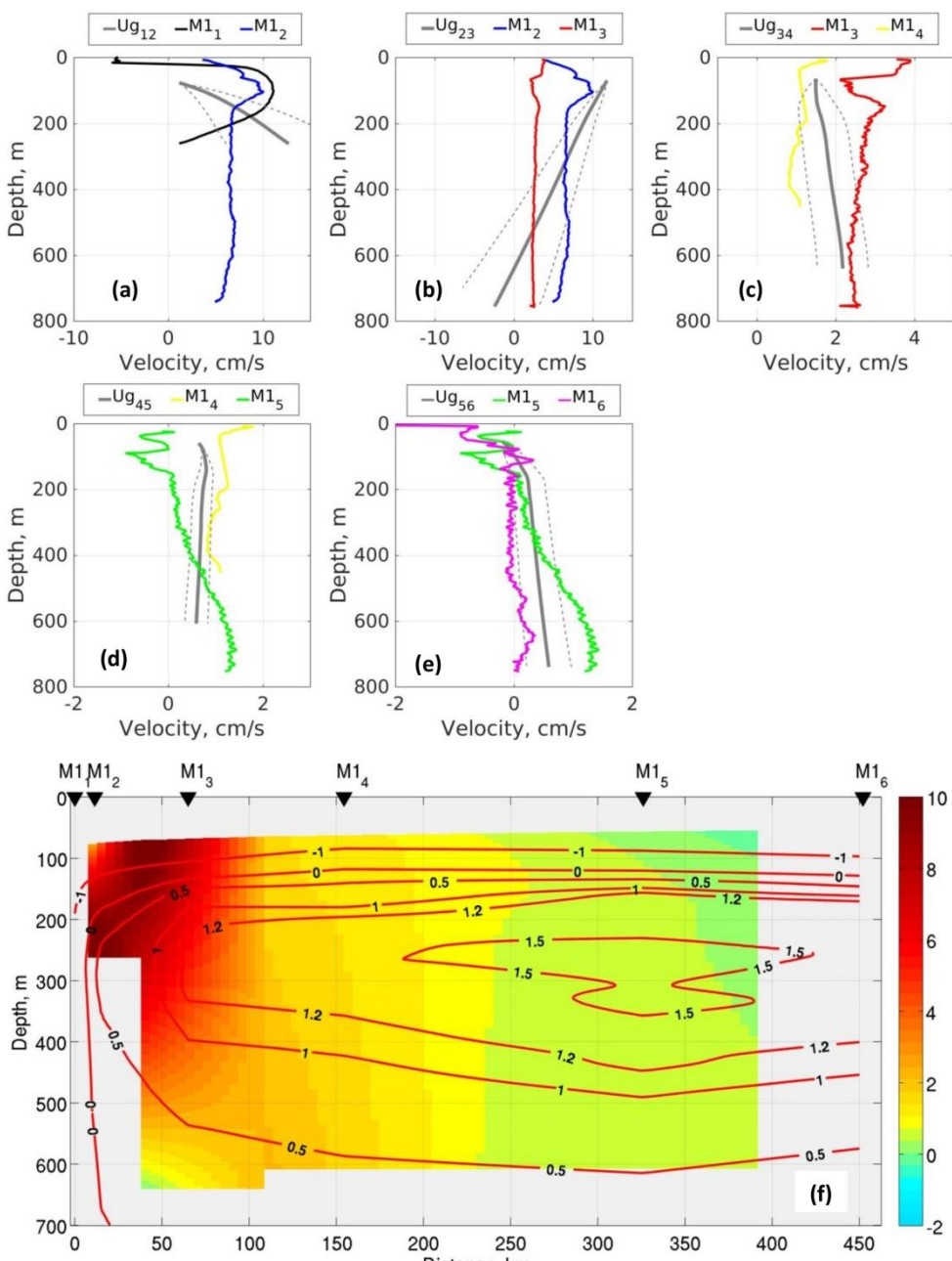

**Figure 6: (a)–(e) Vertical profiles of geostrophic velocities ($u_g$; gray lines; dashed lines indicate intervals of one standard deviation) for each pair of moorings from the Laptev Sea slope. 2013–15 mean eastward velocities at six moorings are shown by color profiles. (f) Cross-slope distribution of geostrophic velocities (color, cm/s) and water temperature along 125ºE section (isolines, °C). All geostrophic velocities were calculated using 2013–15 mean**

temperature and salinity mooring profiles. Positive velocities are eastward. Black triangles show the mooring positions.

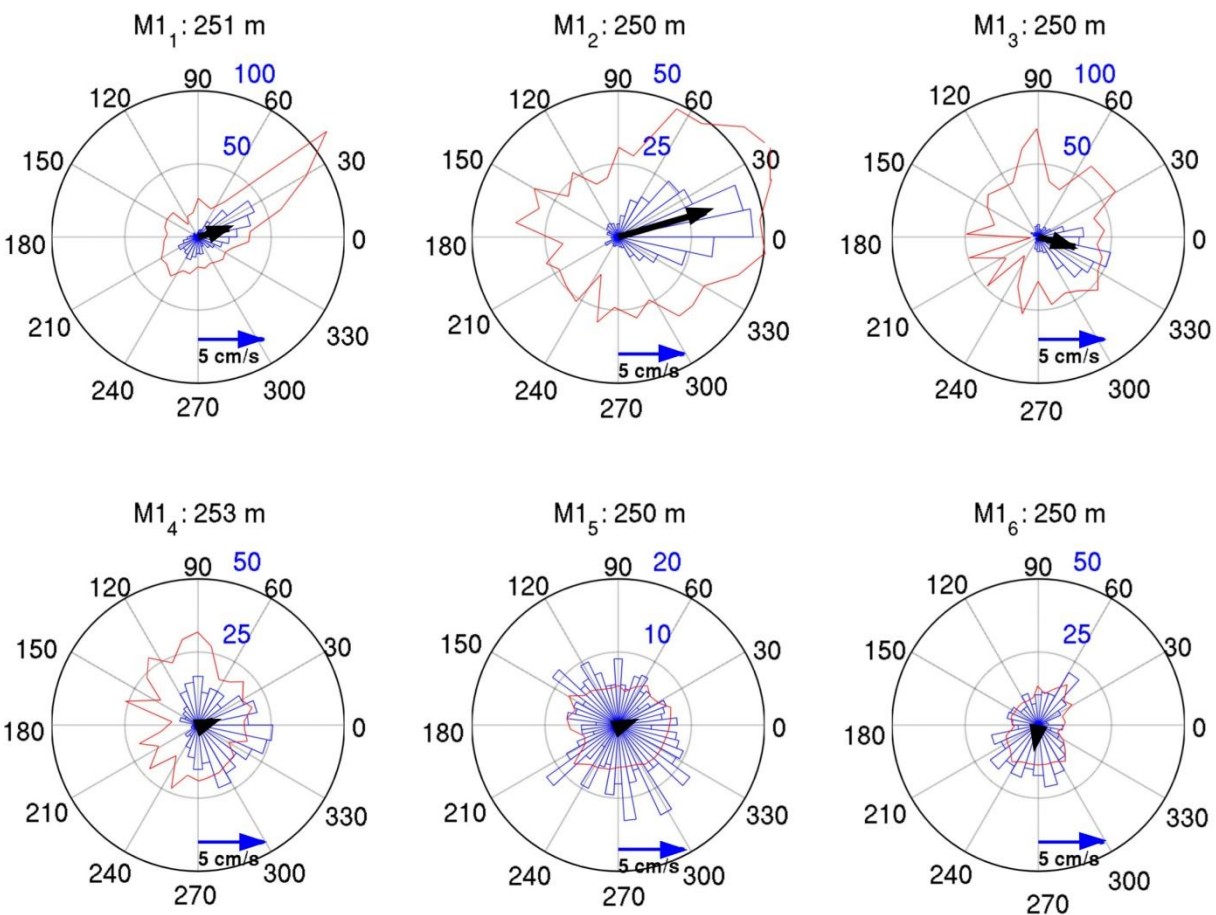

**Figure 7: Current directions and speed at ~250 m (in proximity to AW temperature core) at six moorings at the Laptev Sea slope in 2013–15. Blue sectors show the distribution of current directions (persistence of direction is given in days; gray circles). Red lines indicate the mean current speed within 12º directional bins. The black arrows represent the record-long mean currents. The same linear scale is used to show the mean current speed within directional bins and the mean current vectors.**

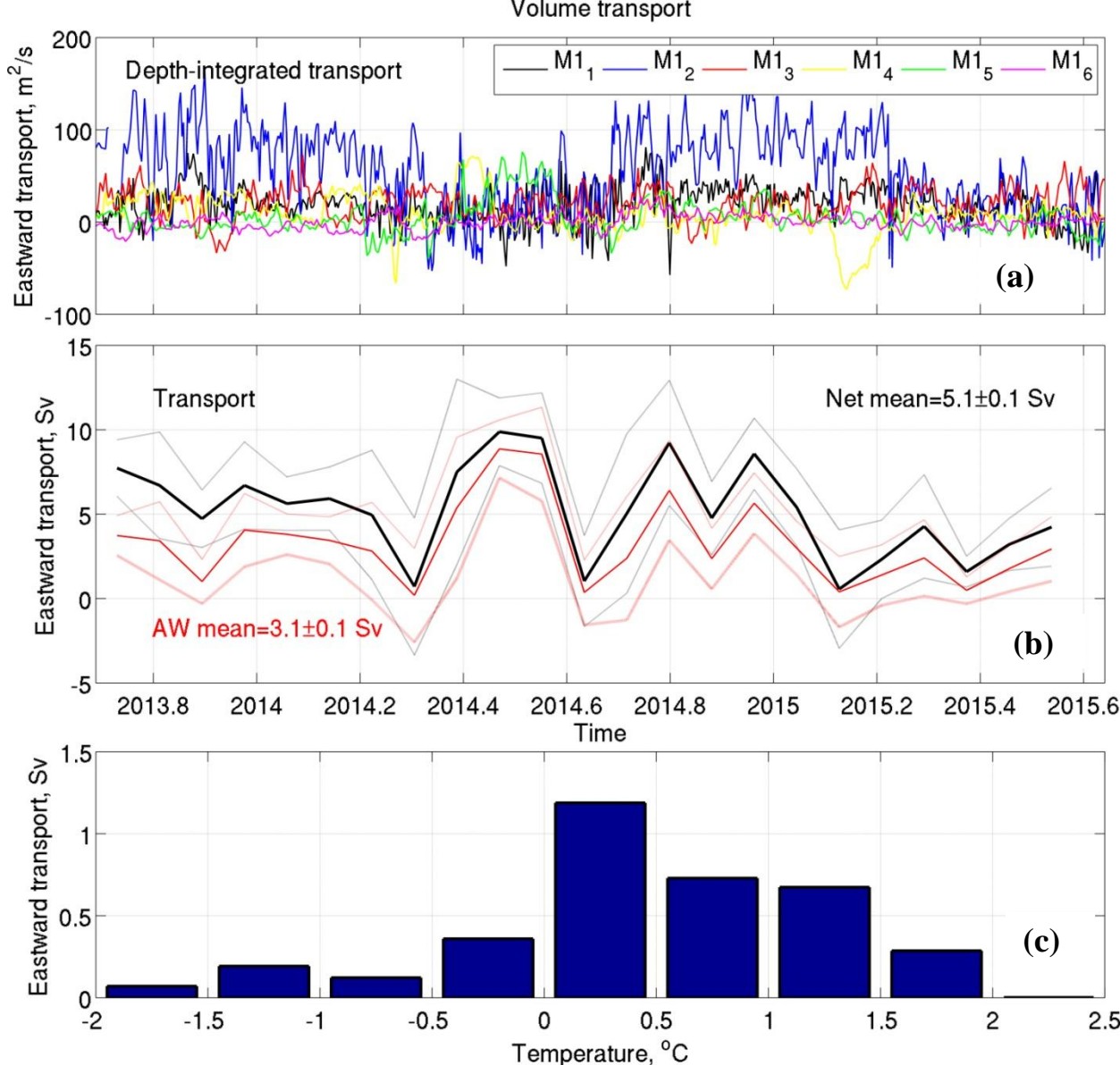

**Figure 8**: **(a) Depth-integrated volume transports (m²/s) in the layer spanned by 2013–15 velocity observations at six moorings at the Laptev Sea slope; (b) monthly net volume transport in the upper 780-m layer (black), and in the AW layer (red) across the 125ºE section. Dotted lines show one standard deviation intervals. (c) Distribution of eastward volume transports averaged over 0.5 ºC temperature bins.**

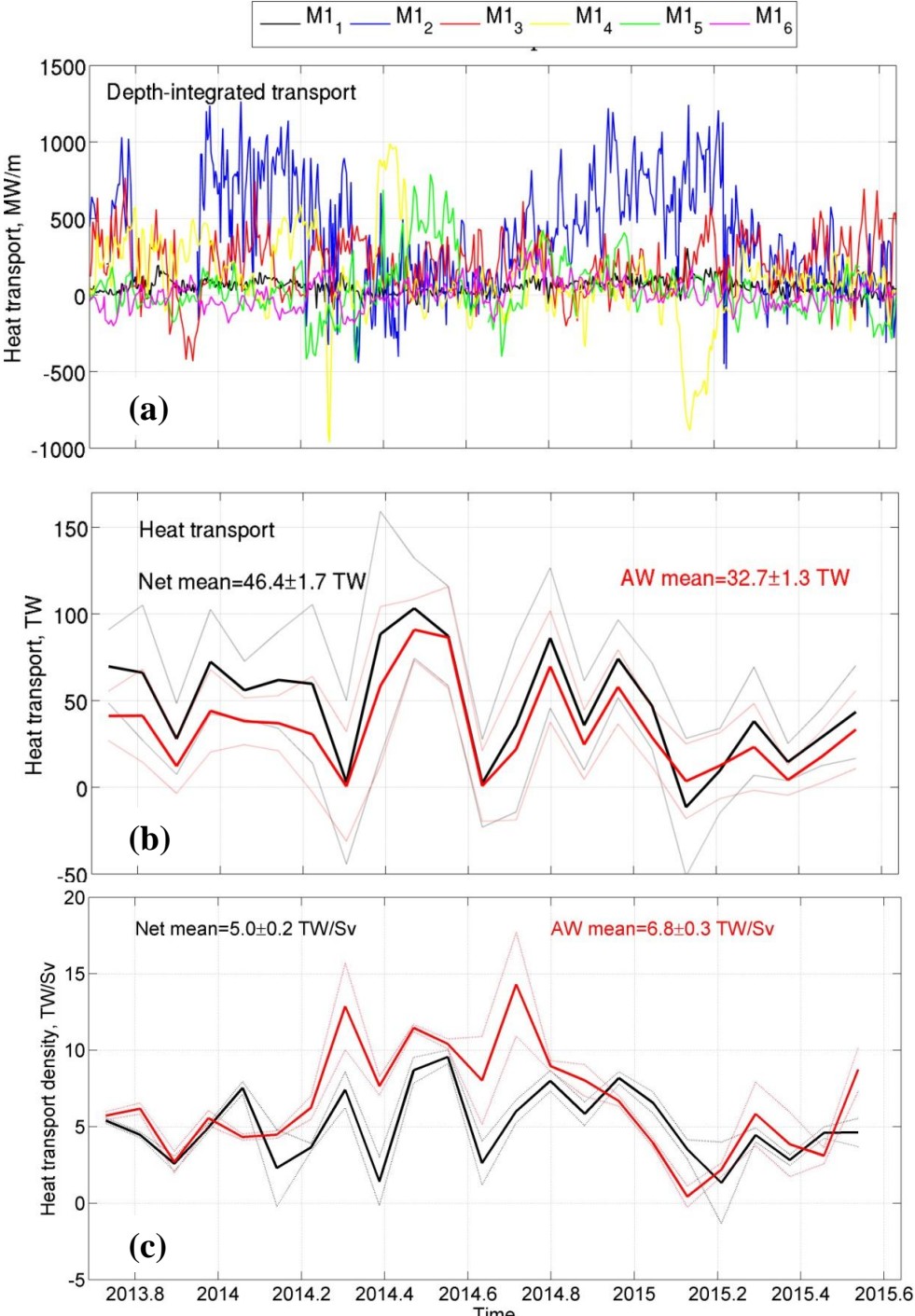

**Figure 9: (a) Depth-integrated heat transports in the layer spanned by 2013–15 CTD and velocity observations at six moorings at the Laptev Sea slope; (b) monthly net heat transport in the upper 780-m layer (black) and in the AW**

layer (red) across the 125ºE section. (c) Monthly density of heat transport across the Laptev Sea slope in the upper 780-m layer (black curve) and in the AW layer (red curve). Dotted lines show one standard deviation intervals.

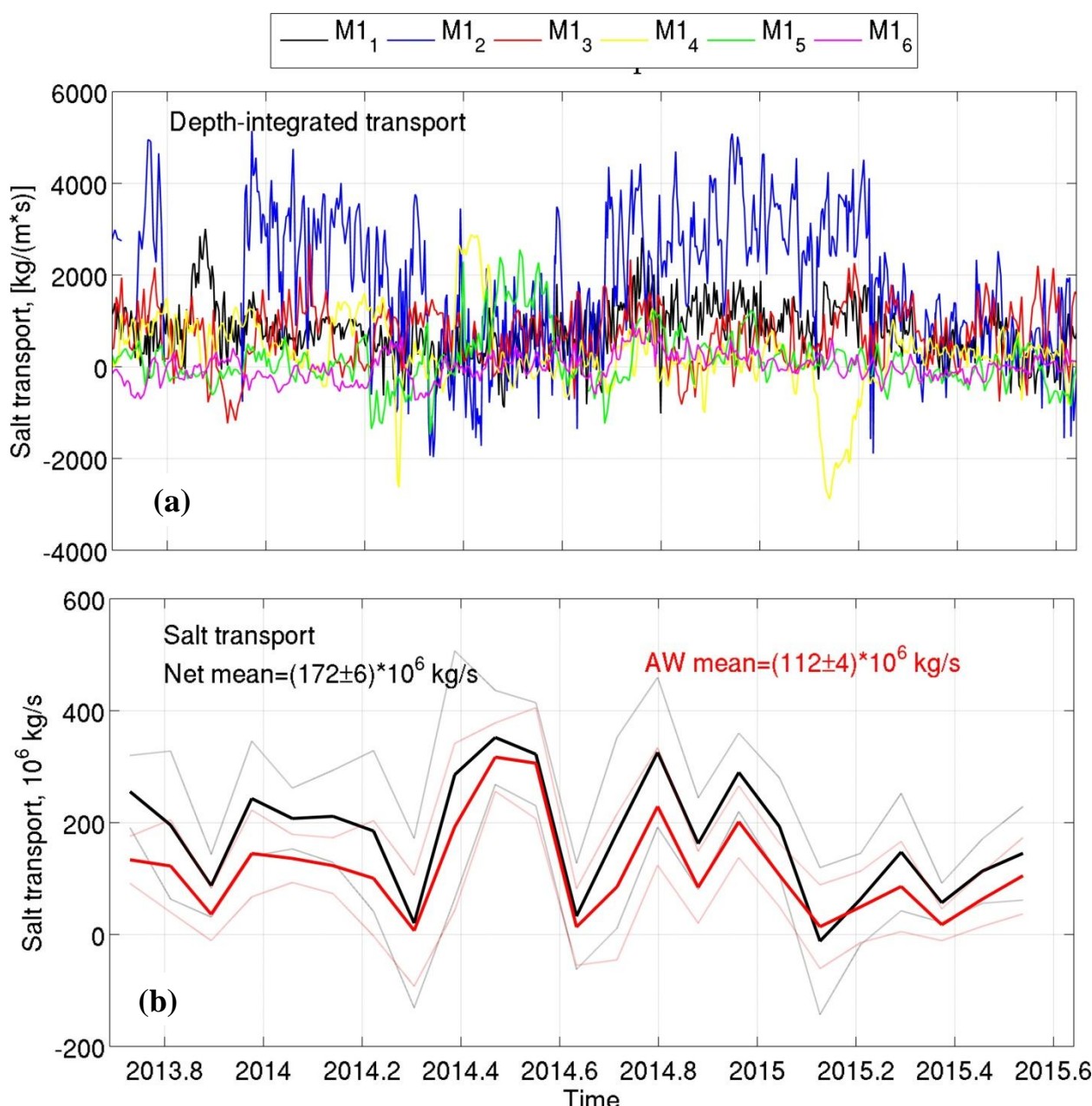

Figure 10: (a) Depth-integrated salt transports in the layer spanned by 2013–15 CTD and velocity observations at six moorings at the Laptev Sea slope; (b) monthly net salt transport in the upper 780-m layer (black) and in the FSAW layer (red) across the 125ºE section. Dotted lines show one standard deviation intervals.

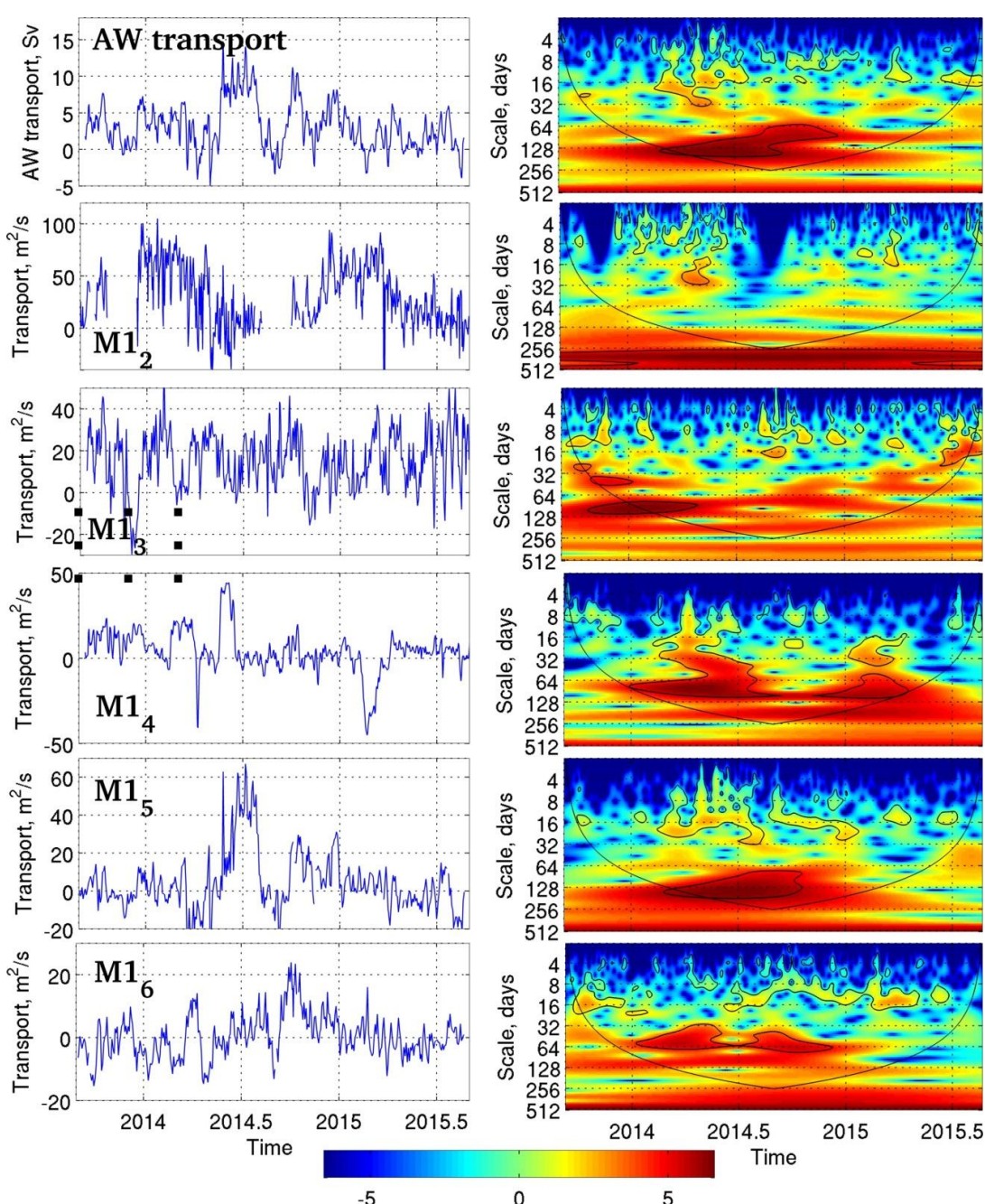

**Figure 11: Depth-integrated AW transports at moorings, and volume transport in the AW layer at the Laptev Sea slope and their wavelet powers (in units of normalized variance). Scales indicate periods defined in wavelet spectral analysis. Black lines show significance at 95 % confidence level. Outside the cone of influence (black lines), the edge effect from finite series length becomes important.**

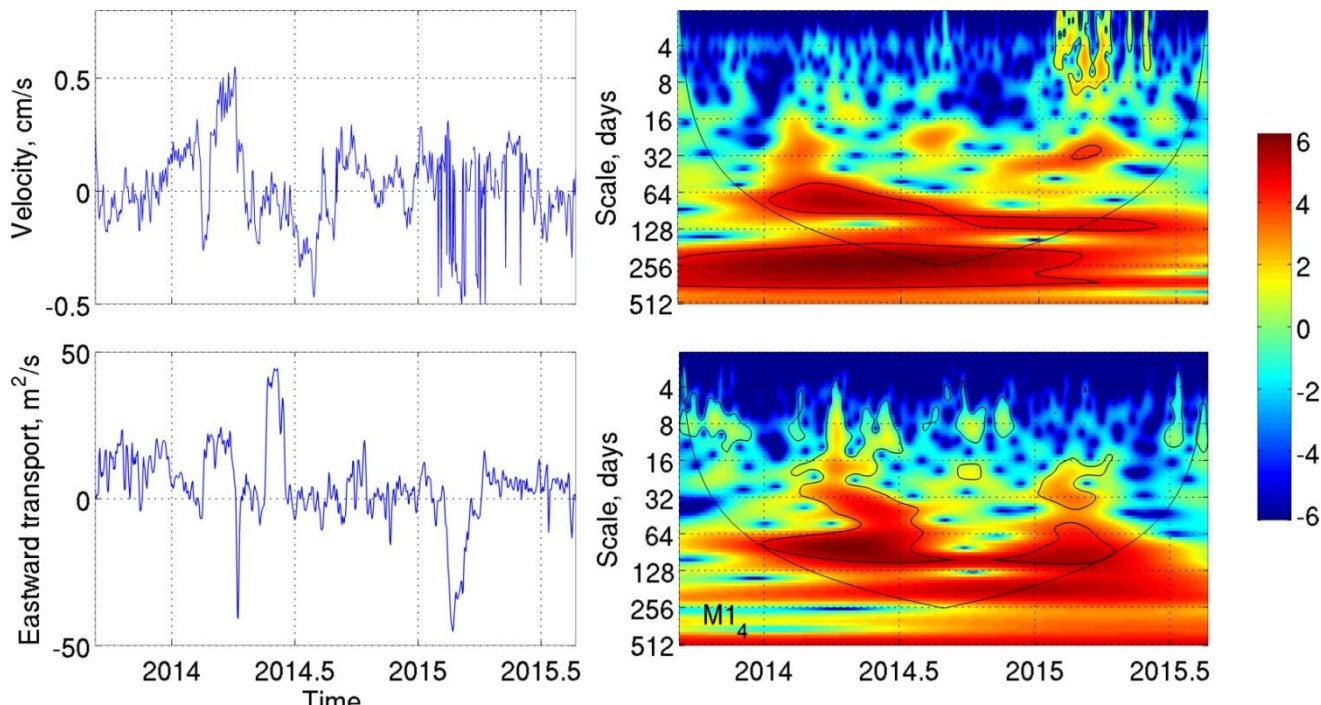

**Figure 12: (Top panels) Depth-averaged baroclinic velocities (cm/s) in the AW layer (positive is eastward) between moorings M1₄ and M1₅, estimated using geostrophic calculations, and their wavelet power. Before averaging, all baroclinic velocities profiles were shifted to satisfy a zero velocity condition at the surface. (Bottom panels) Time-series and wavelet power of depth-integrated transport in the AW layer at mooring M1₄. All wavelet powers are in units of normalized variance. Black lines show significance at 95 % confidence level peaks.**

**Supplementary materials**

**Introduction**

The following supporting information includes a figure to illustrate heat transport at the Laptev Sea slope calculated using the lowest water temperature observed at NABOS moorings in 2013–15 as the reference temperature.

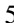

**Figure S1: (a) Depth-integrated heat transports in the layer spanned by 2013–15 CTD and velocity observations at six moorings at the Laptev Sea slope; (b) monthly net heat transport in the upper 780-m layer (black) and in the AW layer (red) across the 125ºE section calculated using the reference temperature of -1.3 ºC (the lowest water temperature observed at NABOS moorings in 2013–15). Dotted lines show one standard deviation intervals.**

