# Peer review of "Heat, salt, and volume transports in the eastern Eurasian Basin of the Arctic Ocean, from two years of mooring observations"

_Ocean Science, 2018_

## Referee Comment (RC1) · M.V Luneva (Referee) · 6 Aug 2018

The paper contains important and solid results based on the analysis of high resolution deep moorings in the Laptev Sea. Estimates of heat and salt transports in the Arctic Ocean are rare, as require expensive services and efforts. Results are new, contain new data of large team of scientists and represent a substantial contribution to understanding water mass transports, pathways and variability in the Arctic Ocean. The most of the paper text is well written and proved. However I recommend to make a major revision, as some editing/rewriting is very desirable and some questions are still open. Major comments: 1. Some sections (1 and 2), especially section 2.2 requires careful

editing and rewriting. This section has a lot of typos (see minor comments below). Being familiar with system of currents in this region, I completely lost understanding after reading this paragraph. I advise to rewrite this paragraph clear, may be to present a table (locations, duration of observations, transports, references of previous studies) and show transects, cited in the text on the figure with all available previous estimates of transports. It will make clear the place of this study in the context of available data and publications. 2. Currents are found to be nearly barotropic and have very strong variability with the time periods from 10 to 100 days. Some profiles are depth intensifying (M15,M16). The authors refer variability to wind forcing and eddies, but don't consider another important player in the Arctic Ocean: topographically trapped barotropic Rossby waves, detected and examined in the Canada basin (Timmermans et al, 2010, J Mar Res). To my mind, It will be useful to check this hypothesis for this part of the Arctic ocean too. Or make a comment, why it is not the case. Scientific significance: 1 Scientific quality : 2 Presentation quality: 2 Some minor comments are below. Page 2: 15. "Observations will be used" or already used? 10. "Branches converges and propagate" . These flows are converges laterally? Which is closer to the shelf break? 14. "We do not have yet such estimates" – may be somebody already has? Better to use "to our knowledge, " Page 3. Paragraph 2.2 is written unclear with a lot of typos. Also you define AW as waters warmer 0C, then "Approximately $3.0 \pm 0.2$ Sv of this transport constitutes AW (water warmer than 2 $^\circ$C) transport into the Arctic" N 25 May be to start with the definitions of AW first and discuss contradictions in definitions? ". In this transport, about $1.3 \pm 0.1$ Sv is carried by the steady Svalbard branch of the WSC (annual mean transports vary in the range of 1.0-1.5 Sv only), whereas $\sim 1.7 \pm 0.1$ Sv is transported by the highly variable (annual mean transports vary in the range of 1-2 Sv) offshore WSC branch—the branch which feeds the flow toward the Yermak Plateau". Please, make this statement clear, may be include diagram. It will be confusing for the reader that "mean net northward transport by the West Spitsbergen Current (WSC) as high as $6.6 \pm 0.4$ Sv", then you talk about mostly eastward transports in the Laptev sea. May be change it by 'alongslope' cyclonic current? 5. "Below the CHL,

both temperature and salinity increase with depth, forming the permanent pycnocline".
It is not quite correct statement, as contradicts statements starting 10. 19-20 a lot of
typos. Page 4: N19: "single moorings, these are" Page 6. "This dataset was used
successfully, for example, in previous studies of long-term changes of the thermoha-
line state of the EB" – to my mind, "successful" is not scientific terminology. Page 7.
The following looks confusing: to use word 'transport' both for depth integrated and
along slope (which is also depth integrated). Is it accepted general terminology? May
be better to use 'depth integrated flux' for the first and 'transport' for along slope value
which is also depth integrated. Use of 'T' for transports and temperature a little bit con-
fusing. May be to change notation? Page 19 N 28. "we consulted simulations". Please,
check this statement. My impression that you could consult somebody, not something.
5.2 page 15. N 4. "We estimated net volume transports for AW using temperature and
salinity measurements from the mooring array" – velocity is missed N8: "This difference
is due to the decrease in mean eastward velocities with depth, so that in the AW layer
velocities become smaller than those observed in the cold halocline and surface mixed
layers." I see strong increase in stations M15 and M16 towards depth. And negative
currents (anti-cyclonic) in the surface layer. How to explain this?

---

## Referee Comment (RC2) · B Rabe (Referee) · 12 Aug 2018

Please see my review in the PDF file (popup comments and comment summary after page 45 – please use each as you see fit). It is here given as the file "...supplement...pdf" Note that my verdict is "publish subject to modest revisions", indicating that these are not major (i.e. are not likely to change the overall conclusions" but are more substantial than just a few minor edits.

Please also note the supplement to this comment:
https://www.ocean-sci-discuss.net/os-2018-36/os-2018-36-RC2-supplement.pdf

---

## Author Comment (AC1) · 13 Sep 2018

We would like to submit the revised manuscript of our paper "Heat, salt, and volume transports in the eastern Eurasian Basin of the Arctic Ocean, from two years of mooring observations" by A. Pnyushkov and co-authors. Attached are the manuscript and our detailed answers to Reviewers' comments. We have modified the text (changes are marked with blue color) responding to the Reviewers' concerns.

Please also note the supplement to this comment:
https://www.ocean-sci-discuss.net/os-2018-36/os-2018-36-AC1-supplement.pdf

---

## Author Response (AR1)

**Reviewer#1**

The paper contains important and solid results based on the analysis of high resolution deep moorings in the Laptev Sea. Estimates of heat and salt transports in the Arctic Ocean are rare, as require expensive services and efforts. Results are new, contain new data of large team of scientists and represent a substantial contribution to understanding water mass transports, pathways and variability in the Arctic Ocean. The most of the paper text is well written and proved.
**We appreciate this assessment.**

However I recommend to make a major revision, as some editing/rewriting is very desirable and some questions are still open.
**Please, find our detailed answers below.**

Major comments:
**Q1**. Some sections (1 and 2), especially section 2.2 requires careful editing and rewriting. This section has a lot of typos (see minor comments below). Being familiar with system of currents in this region, I completely lost understanding after reading this paragraph. I advise to rewrite this paragraph clear, may be to present a table (locations, duration of observations, transports, references of previous studies) and show transects, cited in the text on the figure with all available previous estimates of transports. It will make clear the place of this study in the context of available data and publications.
**A:** We have substantially modified *Section 2* to make it more transparent for the readers. Specifically, we noted that the inflow In Fram Strait includes several types of the AW: the warm AW (water warmer than 2 °C), which propagates as a continuation of the Norwegian Atlantic Current and occupies the upper 400-m layer, and the modified and return AW. In Figure 1, we have added locations of the hydrography sections mentioned in the paper for which we provided the estimates of volume transport. Those estimates have been summarized in a new table (Table 1) as suggested.

**Q2:** p2. Currents are found to be nearly barotropic and have very strong variability with the time periods from 10 to 100 days. Some profiles are depth intensifying (M15,M16). The authors refer variability to wind forcing and eddies, but don't consider another important player in the Arctic Ocean: topographically trapped barotropic Rossby waves, detected and examined in the Canada basin (Timmermans et al, 2010, J Mar Res). To my mind, It will be useful to check this hypothesis for this part of the Arctic ocean too. Or make a comment, why it is not the case.
**A:** We agree with Reviewer's point that barotropic Rossby waves may be a contributor to the variability of currents in the Arctic Ocean at subinertial frequencies. The presence of these waves over the Eurasian slope was confirmed in several past studies (e.g., Voinov and Zakharchuk [1999]; Zakharchuk [2009]). However, our observations do not suggest that these waves play a substantial role in current dynamics at the NABOS moorings. To assess this, we calculated

vertical isopycnal displacement at the M1$_3$ mooring (the mooring located at the steep segment of the continental slope) using the 2013-15 MMP record (**Fig. 1**). Further, we calculated wavelet spectra of the isopycnal displacement at several levels (from 100 through 600 m). In contrast to the wavelet pattern of Rossby waves found, for example, in the deep layer in the Beaufort Gyre (see Fig. 9 in Timmermans et al., [2010]), the calculated wavelet spectra at the M1$_3$ mooring do not show persistent spectral powers at the periods from 7 to 60 days (the typical periods for topographic Rossby waves). Moreover, calculated cross-wavelet spectra at the M1$_3$ suggest low coherence between the vertical isopycnal displacement and lateral velocity components (**Fig. 2**). This low coherence which is unlikely in the presence of Rossby waves for which velocities and isopycnal displacement are closely interrelated.

[Figure]

**Figure1:** (upper panel) Vertical isopycnal displacement (m) and (lower four panels) wavelet transforms of the isopycnal displacement at 100, 250, 400, and 600 m levels at mooring M1$_3$ at the continental slope of the Laptev Sea in 2013-15. All wavelet powers are in units of normalized variance. Black lines show significance at 95 % confidence level. Outside the cone of influence (black lines), the edge effect from finite series length becomes important.

[Figure]

**Figure 2:** Cross-wavelet transforms of the isopycnal displacement and eastward (left panels) and northward (right panels) velocities at 100, 250, 400, and 600 m levels at mooring M1$_3$ at the continental slope of the Laptev Sea in 2013-15. All wavelet powers are in units of normalized variance. Black lines show significance at 95 % confidence level. Outside the cone of influence (black lines), the edge effect from finite series length becomes important.

We have added the essence of this discussion into Section 4.3.

**Q3:** Page 2: 15. "Observations will be used" or already used?
A: In this sentence, we have replaced "..will be used..." with "...were used…".

**Q4:** "Branches converges and propagate" . These flows are converges laterally? Which is closer to the shelf break?
A: We have clarified in the revised text that after the confluence of these two branches, the Barents Sea AW branch flows eastward along the upper part of the EB slope, while the Fram Strait branch occupies a broad segment of the lower slope.

**Q5:** "We do not have yet such estimates" – may be somebody already has? Better to use "to our knowledge, "
A: We have rephrased this sentence as suggested.

**Q6:** Page 3. Paragraph 2.2 is written unclear with a lot of typos. Also you define AW as waters warmer 0C, then "Approximately 3.0±0.2 Sv of this transport constitutes AW (water warmer than 2∘C) transport into the Arctic" N 25 May be to start with the definitions of AW first and discuss contradictions in definitions? ". In this transport, about 1.3±0.1 Sv is carried by the steady Svalbard branch of the WSC (annual mean transports vary in the range of 1.0-1.5 Sv only), whereas ～1.7±0.1 Sv is transported by the highly variable (annual mean transports vary in the range of 1-2 Sv) offshore WSC branch-The branch which feeds the flow toward the Yermak Plateau". Please, make this statement clear, may be include diagram. It will be confusing for the reader that "mean net northward transport by the West Spitsbergen Current (WSC) as high as 6.6 ±0.4 Sv", then you talk about mostly eastward transports in the Laptev sea. May be change it by 'alongslope' cyclonic current?

A: We have substantially modified *Section 2* to make it more transparent for the readers. Specifically, we noted that the inflow In Fram Strait includes several types of AW: the warm AW (water warmer than 2 °C), which propagates as a continuation of the Norwegian Atlantic Current and occupies the upper 400-m layer, and the modified and return AW. In the new Figure 1, we have added locations of the hydrography sections mentioned in the paper for which we provided the estimates of volume transport. Those estimates have been summarized in a new table (Table 1) as suggested.

**Q7:** p5. "Below the CHL, both temperature and salinity increase with depth, forming the permanent pycnocline". It is not quite correct statement, as contradicts statements starting 10.
A: We have modified this sentence pointing out that between the CHL and the AW temperature core, both temperature and salinity increase with depth.

**Q8:** 19-20 a lot of typos.
A: We have corrected the typos in this sentence. Thank you.

**Q9:** Page 4: N19: "single moorings, these are" Page 6. "This dataset was used successfully, for example, in previous studies of long-term changes of the thermohaline state of the EB" – to my mind, "successful" is not scientific terminology.
A: We have removed this word.

**Q10:** Page 7: The following looks confusing: to use word 'transport' both for depth integrated and along slope (which is also depth integrated). Is it accepted general terminology? May be better to use 'depth integrated flux' for the first and 'transport' for along slope value which is also depth integrated. Use of 'T' for transports and temperature a little bit confusing. May be to change notation?
A: The terminology used for depth-integrated and along-slope transports is quite common and implemented widely in literature (e.g., Csanady, *J. Phys. Oceanogr.*, 1973; Webb et al., *EOS* 1991; Masumoto and Yamagata, *J. Geophys. Res.*, 1996; Qu , *J. Phys. Oceanogr.*, 2008 and

many others). Following Reviewer's suggestion, we have replaced all notations for depth-integrated transports throughout the text to avoid potential misinterpretation with temperatures.

**Q11:** Page 19 #28. "we consulted simulations". Please, check this statement. My impression that you could consult somebody, not something.
**A:** We have changed this sentence to "we used simulations performed with...".

**Q12:** page 15 #4. "We estimated net volume transports for AW using temperature and salinity measurements from the mooring array" – velocity is missed
**A:** We have added "velocity measurements" in this sentence.

**Q13:** page 15 #8: "This difference is due to the decrease in mean eastward velocities with depth, so that in the AW layer velocities become smaller than those observed in the cold halocline and surface mixed layers." I see strong increase in stations M15 and M16 towards depth. And negative currents (anti-cyclonic) in the surface layer. How to explain this?
**A:** The noted difference between the net and AW transports also depends on seasonal changes of the deep AW boundary. At the $M1_6$ mooring, this boundary (identified using the position of a 0 °C isotherm) varies from 716 m to the deepest level with temperature measurements, so that the strongest eastward velocities found at the $M1_6$ in the deep layer were not included in the calculation of AW transports. At the same time, these strong velocities were taken into account in estimates of the net transports. We have added this explanation to the text.

**Reviewer#2**

My name is Benjamin Rabe and I was asked to review the manuscript "Heat, salt and volume transports in the Eastern Eurasian Basin of the Arctic Ocean, from two years of mooring observations" by Pnyushkov et al. The manuscript under review for publication in Ocean Science treats an important topic in Arctic physical oceanography, the transports of volume, warm water and freshwater within the Arctic boundary current in the context of large-scale circulation and forcing. Whereas the overall topic is of prime importance in a changing Arctic and the time series analysis methods are sounds there are several issues with the way transports of warm water (referred to as "heat transports") are calculated and reported. In addition there are several smaller details of instrument descriptions and equations that need attention. Overall I recommend this manuscript to be published in Ocean Science subject to modest corrections outlined in my review. My comments are given both as popup comments and PDF comments summaries (two different files).

**We thank Dr. Rabe for his thoughtful reading of our manuscript and very constructive and helpful comments. Below are our responses to the Reviewer's comments.**

**Page 1 Q3, Q4, Q5; Page 2 Q1; Page 7 Q2, Q6**

This is not a heat transport, as volume trasport is not conserved. You can call it a temperature transport and then explain by the known circulation/water masses etc. why this is a useful quantity to consider. You have done so to some extent in Section 3. See my comments there.

**A:** Following notation coming from classic fluid dynamics, the use of the term "temperature transport" assumes a product of temperature and velocity. This is not the quantity we study. Our notation "heat transport" is a reflection of that (similar to salt transport). This notation is well accepted in the oceanographic literature (e.g., Woodgate et al., 2006; 2010; Johns et al., 2011; Li et al., 2017). Thus, we prefer to use the commonly accepted notation for this quantity. Moreover, the temperature transport has an unclear physical meaning in the case of zero-degree water temperatures, so that advection of water with those temperatures has zero temperature transport but enable releasing heat when cooled to the freezing point, for example. We think that our terms for heat and salt transports are acceptable as far as the readers are informed about their physical meanings and warned about possible uncertainties and sensitivities of these estimates in the case of unclosed mass balance. In our revision, we have stressed that point in Section 3 and noted that our estimates of heat and salt transports are valid only for the specific volume of water advected though the Laptev Sea section in 2013-15. In connection to the concern of the unclosed mass balance, we also have pointed out in several places that the estimates of heat and salt transports provided may be sensitive to our choice of reference temperature and salinity.

References:

Woodgate, R. A., K. Aagaard, and T. J. Weingartner (2006), Interannual changes in the Bering Strait fluxes of volume, heat and freshwater between 1991 and 2004, *Geophys. Res. Lett.*, 33, L15609, doi: 10.1029/2006GL026931.

Woodgate, R. A., T. Weingartner, and R. Lindsay (2010), The 2007 Bering Strait oceanic heat flux and anomalous Arctic sea- ice retreat, *Geophys. Res. Lett.*, 37, L01602, doi: 10.1029/2009GL041621.

Johns, W.E., M.O. Baringer, L.M. Beal, S.A. Cunningham, T. Kanzow, H.L. Bryden, J.J. Hirschi, J. Marotzke, C.S. Meinen, B. Shaw, and R. Curry, 2011: Continuous, Array-Based Estimates of Atlantic Ocean Heat Transport at 26.5°N. J. Climate, 24, 2429–2449, https://doi.org/10.1175/2010JCLI3997.1

Li, F., M.S. Lozier, and W.E. Johns, 2017: Calculating the Meridional Volume, Heat, and Freshwater Transports from an Observing System in the Subpolar North Atlantic: Observing System Simulation Experiment. J. Atmos. Oceanic Technol., 34, 1483–1500, https://doi.org/10.1175/JTECH-D-16-0247.1

**Page 5**
**Q1:** To my knowledge, the FSI ACM is a travel-time (actually phase-difference) based velocity sensor, i.e. not using any doppler-effect. However, I have not used this instrument, so I may be wrong. Please check…
**A:** We have changed the text and pointed out that the ACM sensor uses phase differences between acoustic signals to estimate current velocities.

**Page 6**
**Q1:** Did you estimate and include this error in your analysis (e.g. the transports you calculated)?
**A:** We have noted in the text that these errors are individual for each instrument and cannot be quantified without concurrent (non-magnetic) measurements of current directions. Unfortunately, our moorings were not equipped with instruments which could measure current directions that way, and, thus, we cannot provide more reliable estimates of the compass errors.

**Q2:** Would you consider that the best interpolation method? This may warrant an additional sentence or two, at least. There are other methods used in physical oceanography, e.g. the one described by R.F. Reiniger and C.K. Ross, 1968. A method of interpolation with application to oceanographic data. Deep Sea Research, 15, 185-193.
and **Page: 8 Q1:** is this the best interpolation method to choose here? see my comment above…
**A:** The vertical interpolation method does not have a large effect on transport estimates. For comparison, we performed vertical interpolation of lateral velocities at moorings using Reiniger-Ross interpolation method (Reiniger and Ross, 1968). We used these interpolated velocities to estimate depth-integrated volume transports. The relative differences of depth-integrated volume transports with those estimated using the linear interpolation vary in the range from 1.3 to 16%. However, to avoid producing false extrema when interpolating non-monotonic and highly variable with depth velocity components (the known feature of Reiniger-Ross interpolation) we used the linear method to fill gaps between instruments. If the reviewer/editor finds it is important to include the above in the manuscript we can do so.

**Page 7**

**Q1:** Practical salinity actually has not units, different to absolute salinity -- you are using Practical Salinity as defined by the Unesco publications in 1983 to be PSS-78.

**A:** We have removed the units for salinity throughout the manuscript, thank you.

**Q3:** how does this related to you compass error, expected from the weak horizontal magnetic field?

**A:** We have noted that these sectors are substantially wider than the reported instrumental accuracy of measurements for current directions, but may be comparable with those due to the weak horizontal magnetic field.

**Q4:** this is certainly a valid approach. a more robust method would be to use an inverse approach, such as that by Losch et al. (2005) -- this would also give an error estimate based on the mismatch between thermal wind and mooring records: Losch, M., D. Sidorenko, and A. Beszczynska-Moller (2005), FEMSECT: An inverse section model based on the finite element method, J. Geophys. Res., 110, C12023, doi:10.1029/2005JC002910.

**A:** We agree with the Reviewer that geostrophic currents can be estimated using different methods, including that one based on the assimilation of high-resolution CTD and velocity measurements. Even though the approach implemented in Losch et al. (2005) has some advantages when compared to calculations of geostrophic velocities based solely on the thermal wind equations, we note that it will not provide substantial improvement in our case. Specifically, a more accurate spatial pattern for the estimated geostrophic currents (the primary benefit of this method) can be achieved only in the case of higher spatial sampling density of temperatures and salinities observations compared to velocities. For 2013-15 NABOS moorings, temperature and salinity measurements at the section were collected at the same locations across the Laptev Sea slope as velocity measurements. To reflect advantages of the FEMSECT method, we have complemented our estimates of the geostrophic currents over the Laptev Sea slope with their standard deviations as measures of confidence intervals (uncertainties) associated with the method utilized (see the revised Fig. 6). In these calculations, we used daily temperature and salinities profiles at moorings to estimate geostrophic currents and their variability.

**Q5:** there is not really any such thing as "depth-integrated transports" -- what you mean is the depth integrated velocity or the total transport across all depths.

**and Page: 40 Q1:** see my comment earlier -- there is no such thing as "depth-integrated transport" -- if you want to really use this term, it warrants a couple of sentences in section 3, explaining what you really mean and why you are using this term.

**A:** The terminology used for depth-integrated and along-slope transports is quite common and implemented widely in literature (e.g., Csanady, *J. Phys. Oceanogr.*, 1973; Webb et al., *EOS* 1991; Masumoto and Yamagata, *J. Geophys. Res.*, 1996; Qu , *J. Phys. Oceanogr.*, 2008 and many others). We have added to Section 3 that all these depth-integrated transports have a simple

physical meaning. For example, in the case of velocity, $D_w$ represents volume transport within the specified depth range through a unit segment of mooring section. Being summed up over the length of the mooring section, Dw provides the net volume transport.

**Page: 8**

**Q2:** I do not understand equation 3 -- what is cp doing here? you want to get out Ts in Kg m-1 s-1, but you are taking a salinity anomaly, multiplied by cp (specific heat capacity ???) and density…

**A:** Thank you for pointing this mistake out. We have removed the specific heat capacity from this equation.

**Q3:** you are not integrating transports -- you are summing the transports along the section.

**A:** In these calculations we performed the summation of products of depth-integrated transports ($D_W$, $D_H$, and $D_S$) and the length of the section between moorings ($\Delta l$), which is mathematically equivalent to integration.

**Page: 9**

**Q1:** again, this depends highly on your choice of reference temperature…

**A:** We have clarified that the number provided may be changed in case of using an alternate $T_{ref}$.

**Q2:** why does that justify your choice of reference temperature? again, if your anomalies are not very sensitive to Tref you may use this approach of calculating a temperature transport and get some useful meaning out of it. The meaning of the mean value is still unclear to me, though.

**A:** We have added that another reason for choosing the freezing temperature as $T_{ref}$ is that the heat content of the SML in winter is limited by this physical boundary.

**Q4:** This reference only suggested the approach for freshwater fluxes, not heat / temperature fluxes. Please change the text accordingly and state/discuss why this approach is also valid for your "heat" transports.

**A:** We have noted that the approach suggested in Carmack et al. (2016) is for freshwater fluxes, but due to its linear nature the suggested relationship is also valid for heat and salt transports.

**Q5:** OK, so this shows that your variability may be fine, but average values of "heat" transport are not meaningful (e.g. as given in your abstract). Further, you are not referring to any closed volume that is affected by your "heat" fluxes -- the approach by Tsubouchi et al. (2012) and Schauer and Beszczynska-Möller (2009) assumed that their fluxes affect some finite volume.

**A:** We have added the clarification that all our estimates of heat and salt transports are valid only for the specific volume of water advected though the Laptev Sea section in 2013-15.

**Page: 10**

**Q1**: As far as I know, PHC is based on observations contained in the EWG climatology for March-May and July-September. They then fit some sort of sine function to that locally to obtain the fields for each month on a uniform grid. The data contained in EWG stops at 1993. Thus "late 1990s" is not really appropriate.

**A:** Some portions of CTD data utilized in the PHC dataset were collected in the Arctic Ocean in the late 90s and even in the early 2000s (e.g., see the description of Bedford dataset). However, we agree with the Reviewer that for the eastern Eurasian Basin we should limit this period to "the early 1990s" because for that particular region the PHC climatology is based mostly on the EWG observations. The text was updated accordingly.

**Page: 14**

**Q1:** where does the error come from? please give details on how you estimated this error (e.g. instrumental, interpolation errors, variability around the mean...).

**A:** We have modified this sentence explaining that these errors are the standard errors of the monthly means.

**Page: 20**

**Q1:** worth mentioning the Barents Sea inflows of AW here. Even though the total volume input through the easter Fram Strait is 3-4 times that of the Barents Sea (through St. Anna Trough) it may still be significant, and much of the Fram Strait input may recirculate before reaching the Laptev Sea --you do mention the Barents Sea inflow at the bottom of this page, so perhaps link these two discussion items with another sentence?

**A:** We have added that at inter-annual time scales, the variability of the ACBC may be dominated by barotropic forces (e.g., advection of potential vorticity with the Barents Sea branch of the AW) as suggested by good agreement between the mean current in the upper 780 m layer with local topography.

**Page: 21**

**Q1:** again, you would save yourself a lot of explaning and discussing if you called these "temperature transports" and state something about their meaning (e.g. in relation to moving heat or warm water around).

**A:** We have added additional explanations of the terms "heat/temperature" transports to Section 3. Specifically, we have noted that in the case of unclosed volume balance when the advected mass is not conserved, the $F_H$ has meaning of temperature and salinity transports calculated relative to the reference values; when the mass transport is balanced the temperature and salinity transports have physical meanings of heat and salt transports.

**Page: 41**

**Q1:** how do the anomalies of this timeseries look for different reference temperatures? could be a figure in the appendix / supplement…

**A:** We have added a new figure to Supplementary materials as an illustration of heat transports calculated using the reference temperature of -1.3 °C (the lowest water temperature observed at NABOS moorings in 2013-15).

---

## Author Response (AR2)

**Review #1**

**We thank the Editor for his thoughtful reading of our manuscript and very helpful comments. Please, find our detailed answers below.**

**Q1**: P.3, L27. We have changed this part of the sentence to "…and the modified returning AW."

**Q2**: P.5, L6. We have replaced dashes for intervals throughout the text.

**Q3**: P.5, L9. We have removed a dash from this sentence.

**Q4**: P.5, L27. We have changed this reference to Nurser and Bacon (2014).

**Q5**: P.5, L9. We have changed this sentence as suggested.

**Q6**: P.7, L2. We have introduced the suggested change.

**Q7**: P.7, L8. We have added a required article.

**Q8**: P.8, L16. We have changed this sentence as suggested. Now it sounds like "The integral of $D_w$ over the length of the mooring section provides the net volume transport."

**Q9**: P.9, L1. We have changed "limitation" to "limitations" in this sentence.

**Q10**: P.9, L17. We have changed "will increase" to "increases" in this sentence.

**Q11**: P.9, L23-27. We have changed this sentence to make it more transparent and correct.

**Q12**: P.10, L1. We have changed this sentence to "A similar dependence on $S_{ref}$ occurs in calculations of freshwater transports (see Tsubouchi et al., 2012; Carmack et al., 2016 for discussion)."

**Q13**: P.10, L1. We have removed "…relative to fresh water" from this sentence. We have also noted in the text that with $S_{ref} = 0$ salt transport has unambiguous physical meaning even for a non-zero net volume transport.

**Q14**: P.10, L21. We have changed the title of this section to "Water mass and flow structures over the Laptev Sea slope in 2013-15".

**Q15**: P.11, L1. We have changed the title of this section to "Water mass structure over the Laptev Sea slope".

**Q16**: P.11, L30; P.13, L4; P.14, L26. We have added required articles.

**Q17**: P.14, L28. We have added the reference to Zakharchuk (2009), who described Rossby waves with those periods.